# A Spiking Heterogeneous Harmonic Resonate-and-Fire State Space Model for Time Series

**Kartikay Agrawal** [1]  **Vaishnavi Nagabhushana** [1]  **Abhijeet Vikram** [2]  **Vedant Sharma** [2]  **Ayon Borthakur** [1]

## Abstract

Spiking neural networks have attracted increasing attention for their energy efficiency, multiplication-free computation, and sparse event-based processing. In parallel, state space models have emerged as a scalable alternative to transformers for long-range sequence modelling by avoiding quadratic dependence on sequence length. We propose here a spiking heterogeneous harmonic resonate-and-fire state space model ($SH^2$RFSSM), a second-order spiking SSM for classification and regression on ultra-long sequences. $SH^2$RFSSM outperforms transformers and first-order SSMs on average while eliminating matrix multiplications, making it highly suitable for resource-constrained applications. Furthermore, we introduce a kernel-based spiking regressor that enables accurate modelling of dependencies in sequences of up to 50k steps. We also observe a reduction in spiking operations and improved performance with heterogeneity and discretisation in harmonic resonate-and-fire neuronal layers. Overall, we evaluate Harmonic Resonate-and Fire layers across 17 diverse datasets, spanning sensors, time series, and classification to long-term forecasting. Our results demonstrate that $SH^2$RFSSM achieves superior long-range modelling capability with energy efficiency, positioning it as a strong candidate for signal processing on resource-constrained devices for human activity recognition, time series classification, and regression.

---

[1]Mehta Family School of Data Science & Artificial Intelligence, IIT Guwahati, Assam, India [2]IISER Pune, Maharashtra, India. Correspondence to: Kartikay Agrawal <a.kartikay@iitg.ac.in>, Ayon Borthakur <ayon.borthakur@iitg.ac.in>.

*Proceedings of the 43rd International Conference on Machine Learning*, Seoul, South Korea. PMLR 306, 2026. Copyright 2026 by the author(s).

## 1. Introduction

Spike-based deep learning has emerged as an ultra-low-power, sparse computing paradigm for efficient AI in recent years. Spike-based neuromorphic hardware such as Loihi (Davies et al., 2021; Shrestha et al., 2024), TrueNorth (Akopyan et al., 2015), and Dynapse (Richter et al., 2024) utilises far lower resources than conventional ANN-based designs. Apparently, most spike-based models (Zhou et al., 2023; 2025; Lee et al., 2025; Shen et al., 2025; Stan & Rhodes, 2024) rely on integrate-and-fire (IF) or leaky IF neurons, which miss key biological traits like oscillations. While the biophysically detailed Hodgkin-Huxley model captures these dynamics, it is computationally inefficient. Driven by these observations, Resonate-and-fire (RF) neurons (Izhikevich, 2001), computationally as light as IF but more expressive, have gained recent attention (Shrestha et al., 2024; Higuchi et al., 2024; Fabre et al., 2025). However, RF neurons remain underexplored for very long sequence modelling. Moreover, the impact of neuronal heterogeneities in RF neurons has not been studied, although multiple such studies on LIF counterparts (Perez-Nieves et al., 2021; Dahmen et al., 2025) have been conducted.

For sequential tasks, transformers are de facto standards (Vaswani et al., 2017), but they suffer from quadratic dependence on sequence length. Alternatives, such as KV caching (Brandon et al., 2024) and memory updates (Behrouz et al., 2026), reduce overhead but still lack the simplicity of RNNs. State space models (SSMs) (Gu et al., 2022; Smith et al., 2023; Gu & Dao, 2024) and their spiking variants (Stan & Rhodes, 2024; Shen et al., 2025; Bal & Sengupta, 2025) bridge this gap by retaining the RNN computational simplicity while being as expressive as Transformers. But current spiking SSMs are unsuitable for very long sequence tasks. LinOSS (Rusch & Rus, 2025), a second-order ANN-based SSM, utilises stable discretisations with diagonal state matrices to achieve state-of-the-art results on long-range tasks. However, it lacks spike-based communication, which is crucial for energy efficiency in neuromorphic hardware (Shrestha et al., 2024; Imam & Cleland, 2020), as demanded by battery-driven wearables. Hence, in this work, we introduce $SH^2$RFSSM, a second-order spiking SSM designed for extremely long-range tasks and energy-efficient edge AI.

**Our core contributions include:** (1) A detailed study on leveraging discretization and heterogeneity in proposed HRF layers, both in RNN (BHRF) and SSM (S$H^2$RFSSM) setups. (2) A fully spike-based second-order SSM (S$H^2$RFSSM), without any computationally expensive GELU and GLU units, (3) Superior accuracy over first-order SSMs on long-sequence classification, with higher energy efficiency than ANN-based second-order SSMs, (4) An extension to regression via a convolving kernel, outperforming first-order SSMs on 50K-length tasks, (5) HRF layers as a replacement to FFTs (SpikHRFSSM) in a state-of-the-art SNN architecture, SpikF (which are computationally expensive) for long sequence forecasting tasks.

## 2. Related Works

**State Space Models:** Structured State Space Models (SSMs), including S4, Mamba, and LRU, have demonstrated state-of-the-art performance on a wide range of sequence modelling tasks, particularly in language modelling (Gu & Dao, 2024) and time series forecasting (Rusch & Rus, 2025). These models leverage the mathematical structure of state space representations to model long-range dependencies efficiently and scalably (Gu et al., 2022; Smith et al., 2023; Orvieto et al., 2023). Recent work shows that neural oscillators, as universal function approximators, naturally capture long-range temporal dependencies and often outperform traditional models (Rusch & Mishra, 2021a;b; Lanthaler et al., 2023). They benefit from numerically stable Implicit-Explicit (IMEX) and Implicit Euler (IM) discretisations, which better handle complex dynamics than standard Explicit Euler methods. LinOSS (Rusch & Rus, 2025) builds on this oscillator-based perspective, proposing a linear implicit oscillator-based state-space model that combines the stability of implicit solvers with the expressivity of neural oscillatory dynamics. This leads to better inductive biases for learning long sequences, especially when capturing slow or multi-scale temporal phenomena. However, LinOSS is not optimal for resource-constrained applications (such as in edge AI) due to its computational inefficiency.

**Spiking Neural Networks:** Spiking Neural Networks (SNNs) form a biologically inspired class of neural models characterised by discrete spike-based communication. The most foundational models include the IF, and LIF neurons. These have been extended to include adaptation mechanisms, such as in Adaptive LIF (ALIF), and more complex dynamics as in the RF (Izhikevich, 2001) and Harmonic Resonate-and-Fire (HRF) (AlKhamissi et al., 2021) neurons. Many ALIF-based recurrent models have been proposed (Yin et al., 2021; Bittar & Garner, 2022; Deckers et al., 2024), with recent works extending adaptation to RF and HRF neurons (AlKhamissi et al., 2021; Higuchi et al., 2024), which have been evaluated on only small sequence

datasets. SNNs are increasingly modelled within the SSM framework for efficient computation, especially on resource-constrained hardware. Fabre et al. (2025) reformulated LIF and RF neurons as SSMs following Bittar & Garner (2022), achieving strong results on neuromorphic benchmarks. Yet, scaling to extremely long-range sequences remains a key challenge. Recent non-adaptive, first-order SSM-based spiking models, such as S4D-/S4-inspired LIF neurons (Stan & Rhodes, 2024; Bal & Sengupta, 2025; Shen et al., 2025) and RF-based variants (Huang et al., 2024; Zhang et al., 2026) perform well on mid-range tasks (up to 16k sequence length). Notably, none have been evaluated on extremely long-range time series or regression benchmarks. While an FFT-based replacement for attention (Wu et al., 2025) has been introduced for long-range forecasting, its reliance on computationally heavy FFTs limits suitability for neuromorphic deployment (Bal & Sengupta, 2025). Meyer et al. (2025) recently validated the efficacy of an S4D-based spiking SSM by implementing it on Loihi 2. Although this design was efficient, it was evaluated on reasonably simple sCIFAR-10 datasets only. Perez-Nieves et al. (2021); Shen et al. (2024) showed that in LIF/IF-based networks, neural heterogeneity variation in neuron hyperparameters can improve classification performance. In S$H^2$RFSSM, we aim to explore this further by focusing on HRF neurons to investigate the combined effects of heterogeneity and discretisation. Most prior studies have relied on explicit discretisation schemes, with very few exploring alternatives. A notable exception is Baronig et al. (2025), which demonstrated improved performance using a symplectic implicit-explicit (IMEX) scheme over conventional adaptive-LIF models (Bittar & Garner, 2022). In this work, we focus on enhancing the expressivity of HRF neurons by integrating them into a fully spike-based state space model (S$H^2$RFSSM in Figure 1) for human activity recognition and long-range interaction tasks. In addition, we replace the FFT with HRF neurons in the SpikF backbone to study regression forecasting, yielding the spikHRFSSM variant.

## 3. Methods

### 3.1. From Resonate-and-Fire Neuron to Harmonic Resonate-and-Fire Neuron

The Resonate-and-Fire (RF) neuron (Izhikevich, 2001) provides a closer approximation to the Hodgkin-Huxley (HH) model than the Leaky Integrate-and-Fire (LIF) neuron by capturing subthreshold resonance through a 2D linear system with complex eigenvalues. This enables frequency-selective oscillatory dynamics absent in LIF but observed in HH neurons. However, as noted by Higuchi et al. (2024), it relies on complex-valued state variables (Appendix A.1). To simplify implementation, the Harmonic RF (HRF) (AlKhamissi et al., 2021; Higuchi et al., 2024)

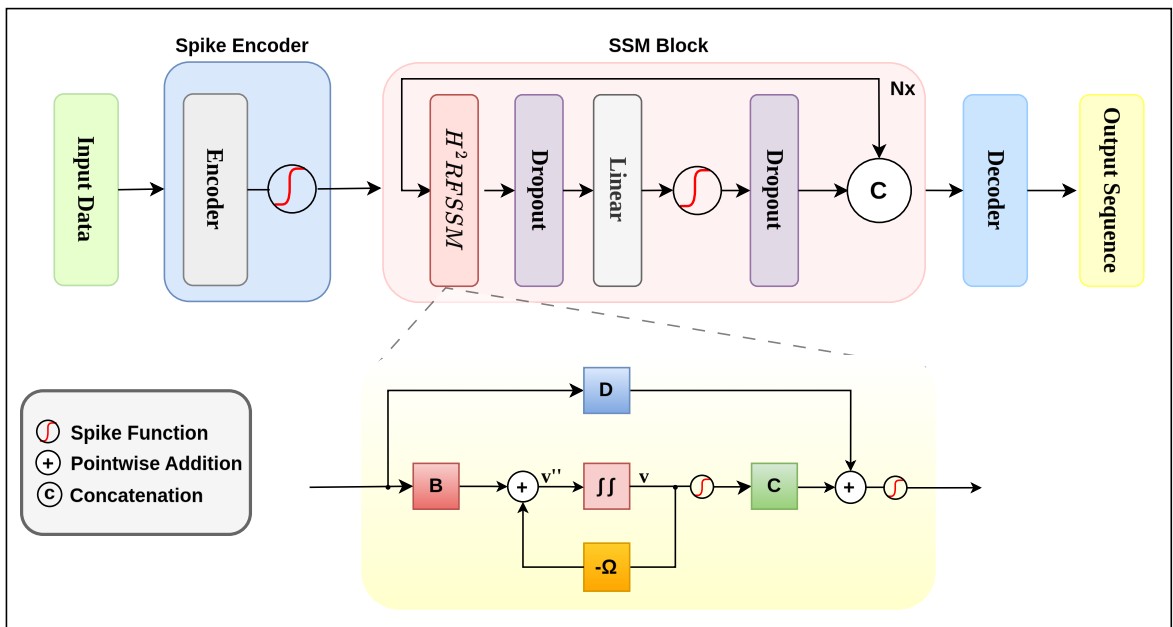

*Figure 1.* Description of S$H^2$RFSSM model. S$H^2$RFSSM consists of a $H^2$RFSSM layer representing multiple states (Eq. 2) followed by a Linear layer repeated for N blocks. Spikes are encoded with a Spike Encoder and decoded using a Spike Decoder.

reformulates RF as a real-valued harmonic oscillator while retaining its resonant dynamics. An HRF neuron is described by (AlKhamissi et al., 2021; Higuchi et al., 2024):

$$u'(t) = -\omega^2 v(t) - 2b\,u(t) + x(t), \qquad v'(t) = u(t). \quad (1)$$

where $v(t)$ denote the hidden state, $u(t)$ as the time derivative of $v(t)$, $x(t)$ is the input spikes to the neuron, $\omega \in \mathbb{R}$ is the frequency parameter and $b \in \mathbb{R}$ is the damping coefficient. However, discretising continuous-time neurons can strongly influence parameter heterogeneity and significantly alter stability. In this work, we evaluate how different discretisation schemes, combined with heterogeneous neuron parameters, affect the behaviour of the HRF neuron

### 3.2. Discretisation and Heterogeneity in HRF Neurons

Oscillatory spiking neurons are sensitive to time discretisation because their dynamics correspond to phase-space rotations. We consider four update schemes: explicit (EX), implicit (IM), and two implicit–explicit variants that differ in how the damping term $b$ is handled (IM b and EX b)(refer to Appendix C.1). We introduce heterogeneity directly into the oscillator dynamics and spiking thresholds. Full discretisation and heterogeneity details are provided in Appendices C.1 and C.2.

### 3.3. Stability Analysis

The choice of the discretisation method significantly affects the stability of an HRF neuron (Figure 2). In the absence of input, the membrane potential converges to a stable rest-

ing equilibrium for values of b and $\omega$ above the respective curve. We examine the effect of step size ($\Delta t$) on the stability properties of different discretisations. For $\Delta t \to 0$, all discretisations remain stable for positive b values. However, the Euler forward (EX) is seen to be unstable for even positive values of the ($\omega$, b) pair as $\Delta t$ increases. Backward Euler (IM) preserves stability for ($\omega$, b) pair where $b < 0$. The dissipative nature of IM leads to sparse spikes, which are favourable for spiking neural networks, as spikes are energy-consuming events. IMEX discretisations consistently exhibit stable behaviour for $b \geq 0$. Hence, we conclude that the choice of discretisation can greatly affect the behaviour of a spiking neural network by design.

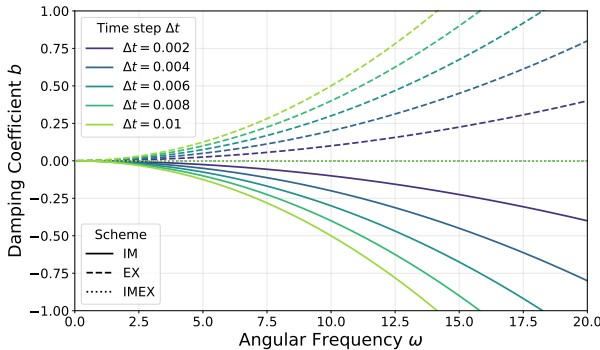

*Figure 2.* Stability Analysis of different discretisations on HRF Neuron with varying time steps. The neuron is stable for ($\omega$, b) values above the line. IMEX (IM b and EX b) exhibit stability in a region similar to the continuous HRF neuron, i.e. $b \geq 0$.

*Table 1.* Impact of heterogeneities and discretisations on HRF neural network. All network types are evaluated on the SHD dataset. Top three models are highlighted in **red**, **blue**, and **violet** respectively.

| Learnable | | | | | BHRF (EX) | | HRF (EX) | | BHRF + $\zeta$ (EX) | | BHRF (IM) | | BHRF (IMEX-B) | |
|---|---|---|---|---|---|---|---|---|---|---|---|---|---|---|
| U | b | $\omega$ | $\theta$ | v | Acc | SOP | Acc | SOP | Acc | SOP | Acc | SOP | Acc | SOP |
| ✗ | ✓ | ✓ | ✗ | ✗ | 92.7±0.6 | 3271.5 | **93.4±0.5** | 3304.6 | 93±0.6 | 3298.9 | 93.2±0.7 | 3162.8 | 92.8±0.3 | 3192.2 |
| ✓ | ✗ | ✓ | ✗ | ✗ | 92.4±0.7 | 3102.2 | 92.5±0.5 | 3161.3 | 93±0.2 | 3148.7 | 93±0.7 | 3078.7 | 93±0.6 | 3148.7 |
| ✓ | ✓ | ✓ | ✗ | ✗ | **93.6±0.3** | 3298.8 | 93.2±0.7 | 3274.1 | 92.7±0.4 | 3284.1 | 93.1±0.5 | 3183.1 | **93.5±0.5** | 3301.1 |
| ✓ | ✓ | ✗ | ✗ | ✓ | 90.8±0.6 | 3279.3 | 90.5±0.4 | 3238.8 | 90.4±0.5 | 3258.9 | 91.7±0.4 | 3008.6 | 91.1±0.7 | 3102.4 |
| ✓ | ✓ | ✓ | ✗ | ✓ | 93±0.3 | 3125.7 | 93.1±1.0 | 3106.8 | **93.4±0.7** | 3093.9 | **93.5±0.2** | **2937.7** | **93.5±0.7** | **2991.1** |
| ✓ | ✓ | ✓ | ✓ | ✗ | 93.3±0.3 | 3280.4 | 93.1±0.5 | 3297.8 | 92.5±0.4 | 3245 | 93.3±0.8 | 3164.2 | **93.4±0.6** | 3164 |
| ✓ | ✓ | ✓ | ✓ | ✓ | **93.6±0.3** | 3107.2 | 93±0.8 | 3056.2 | 93.3±0.4 | 3063.8 | 93±0.5 | **2977.2** | **93.5±0.6** | 3028.1 |

### 3.4. SSM Formulation for HRF neurons

We introduce a learnable input projection $Bx$ and set the damping parameter to $b = 0$. The resulting HRF neuron admits a state-space interpretation, with linearly projected spike inputs and a thresholded readout. Following the stability analysis of the damped HRF dynamics and the characterisation of stable discretisation schemes in the previous section, we now specialise the formulation used in this work.

$$u'(t) = -\Omega v(t) + Bx(t), \qquad v'(t) = u(t),$$
$$y(t) = C\,\Theta\big(v'(t) - \theta_C\big) + Dx(t). \qquad (2)$$

where $v(t), u(t) \in \mathbb{R}^p$ denote the hidden states, $y(t) \in \mathbb{R}^h$ the output, and $x(t) \in \mathbb{R}^h$ the input spike signal. The system is defined by weights $\Omega \in \mathbb{R}^{p \times p}$ which is diagonal, $B \in \mathbb{R}^{p \times h}$, $C \in \mathbb{R}^{h \times p}$, $D \in \mathbb{R}^h$, and an output learnable spiking parameter $\theta_C \in \mathbb{R}^p$. Thresholding $v(t)$ or $u(t)$ is equivalent up to a phase shift and scaling. For the closed-form solution and its phase-based interpretation, refer to Appendix B. Removing explicit damping yields sustained oscillatory dynamics that are effective for long sequences, while stability is ensured by the choice of discretisation. In particular, the implicit (IM) scheme is stable and intrinsically dissipative, accounting for neuronal damping through numerical dissipation, whereas IMEX preserves oscillatory energy in the undamped regime(Rusch & Rus, 2025). We implement the neurons as HRF Layers across different architectures. Let $u_n, v_n \in \mathbb{R}^p$ denote the hidden states and $x_n \in \mathbb{R}^h$ the input spike signal. The discrete-time dynamics are:

$$u_n = u_{n-1} + \Delta t(-\Omega v_\star + Bx_n)$$
$$v_n = v_{n-1} + \Delta t\, u_n \qquad (3)$$

where $v_\star = v_n$ for IM and $v_\star = v_{n-1}$ for IMEX. Spikes are emitted when $v_n \geq \theta_C$, with thresholds $\theta_C \in \mathbb{R}^p$ learned using a step-double Gaussian surrogate gradient (Neftci et al., 2019; Higuchi et al., 2024). As in Higuchi et al. (2024), no reset is applied, enabling parallel-in-time computation.

For understanding the impact of discretisation on an HRF neuron, we conduct a single neuron simulation (Figure 3)

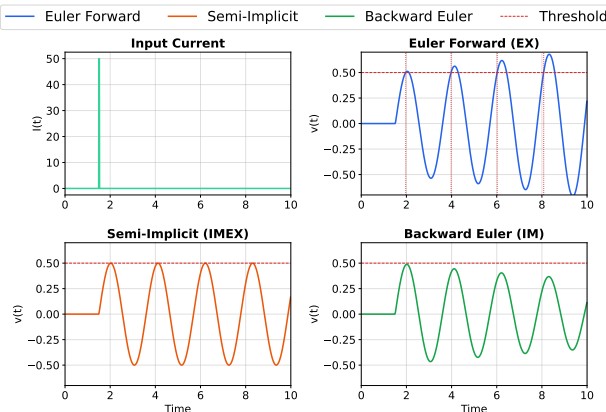

*Figure 3.* Impact of discretisation on HRF Neuron (damp set to 0). Neurons fire upon crossing the threshold and emit spikes. EX spikes frequently whereas IM naturally dampenes the neuron state.

by injecting a unit spike input to an HRF neuron with zero phase. For the continuous case, if the damping parameter 'b' is constrained to be non-negative, we must get stable dynamics. If set to zero, we should observe sustained oscillation. From Figure 3, we can see that for Euler Forward (Explicit), our model is diverging over time, making it unstable. IMEX discretisation is overall stable and hence preserves energy. IM, although dissipative, is still stable. Further phase properties can be refered to in Figure 8.

## 4. Experiments

### 4.1. Heterogeneity on HRF neurons

Heterogeneity is ubiquitous in biology. Recent studies on integrate-and-fire neuron variants have shown that a few combinations of such heterogeneities can also aid learning or improve model performance (Perez-Nieves et al., 2021; Dahmen et al., 2025; Shen et al., 2024). Resonate-and-Fire neurons are more expressive than integrate-and-fire neurons. We here study the impact of heterogeneities as well as discretisations on HRF variants from generalisation and spiking operations (SOPs) / sparsity perspective (Table 1). Higuchi et al. (2024) showed balanced harmonic resonate-

*Table 2.* Best classification performance (mean and standard deviation over five training runs) on UEA time-series datasets using the hyperparameter protocol of Walker et al. (2024). An asterisk (*) denotes results obtained using a different protocol to achieve the best reported performance. For brevity, several dataset names are abbreviated as follows: Worms (EigenWorms), SCP1 (SelfRegulationSCP1), SCP2 (SelfRegulationSCP2), Ethanol (EthanolConcentration), Heartbeat, and Motor (MotorImagery). SNN denotes a spiking neural network. Top four models are highlighted in **red**, **blue**, **violet**, **black** respectively.

| Seq/Class | | 17,984/5 | 896/2 | 1,152/2 | 1,751/4 | 405/2 | 3,000/2 | |
|---|---|---|---|---|---|---|---|---|
| | **SNN** | **Worms** | **SCP1** | **SCP2** | **Ethanol** | **Heartbeat** | **Motor** | **Avg** |
| Transformer Methods | | | | | | | | |
| Transformer (2017) | No | OOM | $84.3 \pm 6.3$ | $49.1 \pm 2.5$ | $40.5 \pm 6.3$ | $70.5 \pm 0.1$ | $50.5 \pm 3.0$ | 59.0 |
| RFormer (2024) | No | $90.3 \pm 0.1$ | $81.2 \pm 2.8$ | $52.3 \pm 3.7$ | $34.7 \pm 4.1$ | $72.5 \pm 0.1$ | $55.8 \pm 6.6$ | 64.5 |
| First Order Methods | | | | | | | | |
| NCDE (2020) | No | $75.0 \pm 3.9$ | $79.8 \pm 5.6$ | $53.0 \pm 2.8$ | $29.9 \pm 6.5$ | $73.9 \pm 2.6$ | $49.5 \pm 2.8$ | 60.2 |
| NRDE (2021) | No | $83.9 \pm 7.3$ | $80.9 \pm 2.5$ | $53.7 \pm 6.9$ | $25.3 \pm 1.8$ | $72.9 \pm 4.8$ | $47.0 \pm 5.7$ | 60.6 |
| Log-NCDE (2024) | No | $85.6 \pm 5.1$ | $83.1 \pm 2.8$ | $53.7 \pm 4.1$ | $34.4 \pm 6.4$ | $75.2 \pm 4.6$ | $53.7 \pm 5.3$ | 64.3 |
| S5 (2023) | No | $81.1 \pm 3.7$ | $89.9 \pm 4.6$ | $50.5 \pm 2.6$ | $24.1 \pm 4.3$ | $77.7 \pm 5.5$ | $47.7 \pm 5.5$ | 61.8 |
| LRU (2023) | No | $87.8 \pm 2.8$ | $82.6 \pm 3.4$ | $51.2 \pm 3.6$ | $21.5 \pm 2.1$ | $78.4 \pm 6.7$ | $48.4 \pm 5.0$ | 61.7 |
| S6 (2024) | No | $85.0 \pm 16.1$ | $82.8 \pm 2.7$ | $49.9 \pm 9.4$ | $26.4 \pm 6.4$ | $76.5 \pm 8.3$ | $51.3 \pm 4.7$ | 62.0 |
| Mamba (2024) | No | $70.9 \pm 15.8$ | $80.7 \pm 1.4$ | $48.2 \pm 3.9$ | $27.9 \pm 4.5$ | $76.2 \pm 3.8$ | $47.7 \pm 4.5$ | 58.6 |
| P-SpikeSSM* (2025) | Yes | $85.5 \pm 7.2$ | $81.9 \pm 2.8$ | $43.1 \pm 8.4$ | $32.2 \pm 6.1$ | $74.4 \pm 7.8$ | $51.3 \pm 7.5$ | 61.4 |
| SpikingSSM* (2025) | Yes | $85.6 \pm 2.3$ | $88.1 \pm 3.2$ | $55.6 \pm 11.6$ | $21.6 \pm 7.4$ | $75.0 \pm 7.9$ | $55.6 \pm 12.4$ | 63.6 |
| AUSSM* (2025) | No | $82.6 \pm 3.4$ | $64.2 \pm 4.9$ | $4.2 \pm 6.8$ | $4.7 \pm 4.1$ | $53.0 \pm 3.81$ | $2.6 \pm 5.5$ | 35.2 |
| PD-SSM (2025) | No | $90.0 \pm 5.7$ | $80.9 \pm 2.0$ | $56.1 \pm 8.6$ | $34.7 \pm 4.0$ | $80.0 \pm 2.6$ | $60.0 \pm 3.7$ | 67.0 |
| Second Order Methods | | | | | | | | |
| BioOSS* (2025) | No | $92.8 \pm 5.2$ | $85.6 \pm 3.9$ | $55.1 \pm 1.8$ | $33.4 \pm 10.7$ | $74.8 \pm 2.0$ | $55.8 \pm 5.8$ | **66.3** |
| LinOSS (2025) | No | $95.0 \pm 4.4$ | $87.8 \pm 2.6$ | $58.9 \pm 8.1$ | $29.9 \pm 0.6$ | $75.8 \pm 3.7$ | $60.0 \pm 7.5$ | **67.9** |
| Lin-HSSM-BPTT (2025) | No | $78.3 \pm 7.5$ | $86.4 \pm 1.8$ | $63.9 \pm 7.3$ | $29.9 \pm 0.6$ | $73.9 \pm 0.6$ | $48.1 \pm 5.7$ | 63.4 |
| Lin-HSSM-RHEL (2025) | No | $75.0 \pm 9.9$ | $86.1 \pm 2.9$ | $61.4 \pm 9.4$ | $29.9 \pm 0.6$ | $73.5 \pm 1.6$ | $51.6 \pm 5.0$ | 62.9 |
| Nonlin-HSSM-BPTT (2025) | No | $51.1 \pm 7.2$ | $86.8 \pm 3.2$ | $54.0 \pm 4.9$ | $29.9 \pm 0.6$ | $74.5 \pm 2.4$ | $56.5 \pm 7.6$ | 58.8 |
| NonLin-HSSM-RHEL (2025) | No | $50.6 \pm 6.7$ | $85.6 \pm 4.4$ | $54.0 \pm 2.0$ | $29.9 \pm 0.6$ | $73.9 \pm 4.3$ | $53.0 \pm 5.7$ | 57.8 |
| S$H^2$RFSSM | Yes | $92.8 \pm 3.3$ | $84.2 \pm 3.2$ | $59.3 \pm 7.7$ | $34.7 \pm 7.1$ | $74.5 \pm 3.4$ | $59.6 \pm 5.4$ | **67.5** |

and-fire neurons achieve high performance on the SHD dataset with an accuracy of $92.7 \pm 0.7$ % and 4139.5 SOPs using a BHRF neuron. We performed a detailed grid search on all heterogeneous neuronal parameters $(u, b, \omega, \theta, \Delta t, v)$ (Refer to Appendix Figure 7). We observe that heterogeneity and discretisation can improve model performance while significantly reducing SOPs. For example, using BHRF with IM discretisation results in a 0.7% improvement in accuracy, with SOPs reduced to 2937.7, corresponding to $1.4\times$ lower SOPs (Table 1). We also observed similar trends in our study in an SSM framework (refer section 4.4). Hence, we conclude that heterogeneity and discretisation affect overall performance not only in terms of accuracy but also in SOPs, which directly translate into energy consumption.

We utilise a vanilla HRF network equivalent to that of Higuchi et al. (2024) and introduce various forms of heterogeneity and discretisation to study their impact on model performance. Following a systematic grid search over all learnable parameter combinations, we select the top seven models to evaluate different discretisation schemes (see Appendix C.3). We find that learning $u$, $b$, and $\omega$ is most beneficial, while a learnable $v$ reduces SOPs. Further, the model's performance is insensitive to discretisations. Similarly, relaxing the BHRF neuron to an HRF neuron, or adding an extra learnable $\zeta$ for the soft reset, doesn't sig-

nificantly degrade the model's performance. However, implicit and implicit–explicit discretisations consistently yield lower SOPs, likely due to more stable neuronal dynamics that produce fewer spikes and consequently lower energy consumption. In the follow-up experiments, we incorporated such HRF neurons in an SSM framework to study very-long-range interactions and Human Activity Recognition (S$H^2$RFSSM in Figure 1) and Multivariate Long-Term Forecasting (SpikHRFSSM).

### 4.2. Very-Long Range Interactions

#### 4.2.1. CLASSIFICATION PERFORMANCE

For classification experiments, we show S$H^2$RFSSM's efficacy on the long-range sequential benchmark introduced in (Walker et al., 2024; Rusch & Rus, 2025), using pre-selected random seeds for dataset splitting (70/15/15 splits) and a fixed model hyperparameter range. To evaluate performance on very long temporal dependencies, we study the PPG-DaLiA dataset (49,920 sequences) for regression. Following the hyperparameter protocol of Rusch & Rus (2025) and employing Bayesian search (Akiba et al., 2019), we ensure an optimised training. We compare our model performance to Transformers (Vaswani et al., 2017; Moreno-Pino et al., 2024), State Space Models(Smith et al., 2023; Gu & Dao,

2024; Rusch & Rus, 2025) and Neural ODEs(Morrill et al., 2021; Kidger et al., 2020; Walker et al., 2024). LinOSS (Rusch & Rus, 2025) leverages second-order ODEs for long-range sequence learning and outperforms other SOTA models on such very long-range benchmark datasets. Similarly, S$H^2$RFSSM, on average, performs better than Mamba & S6 by 8.9% & 5.5% respectively (Table 2) respectively, while being only 0.4% lower than SOTA LinOSS. For the long-range classification task on EigenWorms, we observe that our model performs comparably to LinOSS with only 2.2% drop in performance for IM discretisation, and significantly outperforms the IMEX counterpart by 10% (see Table 8). Moreover, we observe a 69× improvement in energy for EigenWorms (Table 5). Our model outperforms first-order SSMs such as LRU and S6 by 5.8% and 5.5%, and transformer-based methods such as RFormer (Moreno-Pino et al., 2024) by 3% respectively. Vanilla transformer methods (Vaswani et al., 2017) can suffer from out-of-memory issues on a 24GB GPU budget, due to their quadratic complexity of longer sequences. However, shorter sequences can learn better in some cases. Our model is the only second-order SSM to perform as well as first-order methods such as PD-SSM (Terzic et al., 2025) on the Ethanol data set, just behind Transformer methods, with a drop of 5.8%. On the 3k MotorImagery dataset, we observe that S$H^2$RFSSM with Batch Normalisation of membrane potentials achieves 59.6% (vs. 60.0% for PD-SSM & LinoSS). We compared our model against first-order spiking state-space models using their respective best hyperparameter configurations. Our reproduction of their setups shows that our approach achieves a 6.1% improvement over P-SpikeSSM(Bal & Sengupta, 2025) and a 3.9% improvement over SpikingSSM(Shen et al., 2025), while notably avoiding the use of dense matrix multiplications and nonlinear activation functions such as GeLU or GLU that are employed in prior work(Bal & Sengupta, 2025; Stan & Rhodes, 2024). We report a detailed comparison of discretisations for second-order methods in Appendix D.5.2. Overall, our second-order spiking SSM outperforms Transformers, Neural ODEs, and first-order SSMs while being highly energy-efficient.

### 4.2.2. REGRESSION PERFORMANCE

We present the first regression results of a spiking SSM on extremely long sequences (up to 50k). To address the limited output range of spiking neurons, we introduce a kernel-based spiking regressor with a learnable temporal kernel. As shown in Table 3, S$H^2$RFSSM models consistently outperform all first-order SSMs. S$H^2$RFSSM-IMEX surpasses Mamba by 0.022 MSE, demonstrating both the energy efficiency of IMEX discretisation and the representation power of resonating neurons. Despite their efficiency, our models are only at most 0.027 MSE below second-order SSMs. We

observed from Rusch & Rus (2025) that IM discretisation outperforms IMEX for all datasets for classification and vice versa for regression. This trend is also observed in our model. Additionally, Hu et al. (2025) demonstrated that linear SSM-based models outperform RNN-based models for chaotic systems. Pourcel & Ernoult (2025) observed similar trends in the EigenWorms dataset, where the non-linear dynamics from Rusch & Mishra (2021b) suffer a significant drop in performance; however, in the PPG dataset, it outperforms its linear counterpart and only lags behind LinOSS-based models.

*Table 3.* Mean-Squared Error (MSE ×10⁻²) for PPG-DaLiA dataset across five training runs for the best model using hyperparameter protocol from (Walker et al., 2024). SNN implies a spiking neural network. Top three models are highlighted in **red**, **blue**, **violet**, **black** respectively).

| Method | SNN | MSE ↓ |
|---|---|---|
| NCDE (2020) | No | 13.5 ± 0.7 |
| NRDE (2021) | No | 9.9 ± 1.0 |
| Log-NCDE (2024) | No | 9.6 ± 0.6 |
| S5 (2023) | No | 12.6 ± 1.3 |
| LRU (2023) | No | 12.2 ± 0.5 |
| S6 (2024) | No | 12.9 ± 2.1 |
| Mamba (2024) | No | 10.7 ± 2.2 |
| BioOSS* (2025) | No | **7.7 ± 0.2** |
| LinOSS-IM (2025) | No | **7.5 ± 0.5** |
| S$H^2$RFSSM-IM | Yes | 11.8 ± 0.9 |
| RHEL-Lin (2025) | No | 9.5 ± 1.0 |
| RHEL-Nonlin (2025) | No | **8.4 ± 0.5** |
| LinOSS-IMEX (2025) | No | **6.4 ± 0.2** |
| S$H^2$RFSSM-IMEX | Yes | **8.5 ± 0.5** |

### 4.3. Human Activity Recognition

Although our model is designed for long-range sequence datasets, its performance is evaluated on human activity recognition(HAR) datasets. S$H^2$RFSSM achieves 99% accuracy and outperforms the best-performing model (Li et al., 2022) for UCI-HAR by 0.2%. For SHAR, our model achieves 92.7%, which is better than non-spiking models in terms of energy and performance, and is competitive with spiking models, as it outperforms SpikeDeepConvLSTM (Li et al., 2022) by 0.6% and falls behind SpikeDCL by 1.2% (refer to Figure 4). For SHAR, we believe the slight performance drop can be addressed by aligning the spike encoder with the subsequent network.

### 4.4. Heterogeneity on S$H^2$RFSSM and Energy Analysis

Similar to Table 1, we here attempt to understand the energy-efficiency and performance benefits arising from heterogeneity in neuronal parameters and discretisation on S$H^2$RFSSM. Firstly, we compared discretisation and nor-

*Table 4.* Detailed heterogeneity analysis on the UEA benchmark datasets. The learnable $b$ parameter, adapted from Boyer et al. (2025), showed weaker average performance compared to other methods (improving only SCP1) and was therefore excluded from further analysis (see Figure 9 for distribution visualisations of the overall best-performing models). We isolated the effect of heterogeneity using the best-performing configuration (IM discretisation except for Ethanol, and no batch normalisation except for Motor). Experiments with learnable $(u, v)$ and fully homogeneous parameters $(u, b, \Omega, \theta, \Delta t, v)$ were conducted to isolate the gains arising from heterogeneity. The best-performing model on average is reported in Table 2.

| $u$ | $b$ | $\Omega$ | $\theta$ | $\Delta t$ | $v$ | BN | Disc | Worms | SCP1 | SCP2 | Ethanol | Heartbeat | Motor | Average |
|---|---|---|---|---|---|---|---|---|---|---|---|---|---|---|
| ✗ | ✗ | ✓ | ✓ | ✓ | ✗ | ✓ | IM | $85.6 \pm 8.1$ | $82.6 \pm 4.5$ | $55.4 \pm 6.9$ | $30.1 \pm 1.47$ | $74.1 \pm 5.9$ | $\mathbf{59.6 \pm 5.4}$ | 64.4 |
| ✗ | ✗ | ✓ | ✓ | ✓ | ✗ | ✓ | IMEX | $85.6 \pm 2.7$ | $82.6 \pm 4.2$ | $53.0 \pm 4.8$ | $32.4 \pm 4.6$ | $73.2 \pm 5.73$ | $58.6 \pm 7.3$ | 64.2 |
| ✗ | ✓ | ✓ | ✓ | ✓ | ✗ | ✓ | IMEX | $83.3 \pm 4.6$ | $\mathbf{86.8 \pm 2.0}$ | $57.2 \pm 5.2$ | $26.3 \pm 7.0$ | $69.4 \pm 3.8$ | $51.2 \pm 4.4$ | 63.5 |
| ✗ | ✗ | ✓ | ✓ | ✓ | ✗ | ✗ | IM | $\mathbf{92.8 \pm 3.3}$ | $84.2 \pm 3.2$ | $\mathbf{59.3 \pm 7.7}$ | $30.1 \pm 1.47$ | $\mathbf{74.5 \pm 3.4}$ | $58.2 \pm 3.2$ | 66.5 |
| ✗ | ✗ | ✓ | ✓ | ✓ | ✗ | ✗ | IMEX | $90.0 \pm 5.7$ | $82.6 \pm 4.2$ | $53.0 \pm 4.8$ | $34.7 \pm 7.1$ | $73.2 \pm 5.73$ | $58.6 \pm 7.3$ | 65.3 |
| ✗ | ✗ | ✓ | ✓ | ✓ | ✗ | ✗* | IM* | $92.8 \pm 3.3$ | $84.2 \pm 3.2$ | $59.3 \pm 7.7$ | $34.7 \pm 7.1$ | $74.5 \pm 3.4$ | $59.6 \pm 5.4$ | $\mathbf{67.5}$ |
| ✗ | ✗ | ✗ | ✗ | ✗ | ✗ | ✗* | IM* | $75.6 \pm 17.3$ | $79.1 \pm 3.4$ | $52.9 \pm 6.1$ | $31.9 \pm 2.6$ | $75.8 \pm 4.9$ | $55.1 \pm 5.2$ | 61.7 |
| ✓ | ✗ | ✓ | ✓ | ✓ | ✓ | ✗* | IM* | $92.6 \pm 3.5$ | $79.1 \pm 3.5$ | $58.9 \pm 5.2$ | $32.6 \pm 4.8$ | $76.1 \pm 2.1$ | $56.5 \pm 5.1$ | 65.9 |

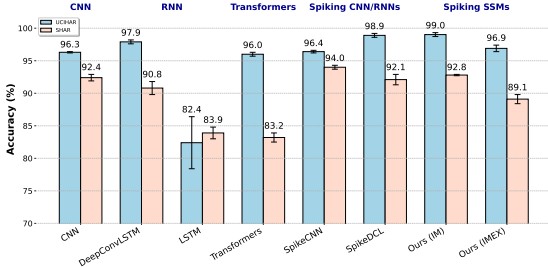

*Figure 4.* Performance comparison (Accuracy %) between different networks on Human Activity Recognition datasets: UCI-HAR(Anguita et al., 2013), and SHAR (Micucci et al., 2017). The evaluation protocol follows the benchmark setup of Li et al. (2022).

malisation combinations in Table 4 (Detailed comparison in Table 8). Then, as per the best performing combinations, we additionally decoupled the impact of heterogeneity on performance in S$H^2$RFSSM. We compare the inference accuracy between a homogeneous and a heterogeneous S$H^2$RFSSM model. Notably, model heterogeneity can be introduced by initialisation alone, by learning alone, or by both. Hence, for a fair homogeneous baseline, u,v, and b are set to zero, $\Delta t$ is set to 0.01, $\theta$ to 1 and $\Omega$ is set with a unity trace. None of them undergoes learning. We observe that a variant of the heterogeneous model always outperforms the homogeneous models. Moreover, heterogeneity across all is not necessarily better than heterogeneity within a subset (65.9% across all vs. 67.5% within a subset). Specifically, we observe that all heterogeneous parameters improved heartbeat performance the most on the smallest dataset in the benchmark. We did not include the $b$ parameter due to the damping nature of IM discretisation and its inefficacy in previously reported results. For SOP-based energy comparison, we observe that spiking rates for the encoders are comparable across IM and IMEX discretisation (Figure 5). However, we observe a reduction in spike rates for $\theta_C$ in IM discretisation. We saw a similar trend in Table 1. We observe an accuracy v/s SOPs tradeoff from Table 5. Learnable u and v can help capture phase informa-

tion and improve overall model performance. However, the models with learnable $b$ may lower overall performance, as observed, and rarely improve it (Table 4). IM discretisation requires fewer steps to converge to optimal accuracy, as we also observed to be a general trend for classification tasks. We summarise our studies on heterogeneity as follows for different parameter learnable regimes:

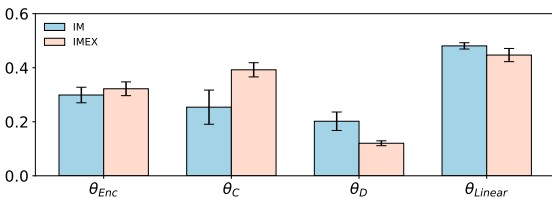

*Figure 5.* Variation in spike rates of S$H^2$RFSSM components across IM and IMEX discretisations. The error bars reflect spike rate variability across the dataset and different blocks.

**For HRF in RNN setup (BHRF):**(1) We observe **no benefit** of using a **learnable $\Delta t$**. (2) $(\mathbf{u}, \mathbf{b}, \omega, \mathbf{v})$ heterogeneity is the most optimal combination such that: (i) IM significantly reduces SOPs to **2937.7** from reported results of **4139.5**. Additionally, there is a **0.7%** improvement in accuracy over the baseline. (ii) **IMEX** resulted in reduced SOPs to **2991.1** but **higher error bars**. This is consistent with classification performance in the SSM setup, where we observed that **IM** performs well on classification tasks. (3) $(\mathbf{u}, \mathbf{b}, \omega, \theta, \mathbf{v})$ heterogeneity results in **reduced SOPs** in general, but **not the most optimal combination**. Hence, learning all parameters is not guaranteed to improve performance.

**For HRF in SSM setup (S$H^2$RFSSM):** (1) Without heterogeneity, though we observe good performance for the smallest length heartbeat dataset, the **homogeneous version** on average only achieves **61.7%**. On the subsequently longer sequence datasets: EigenWorms, SCP1 and SCP2, model **heterogeneity delivers** a significant performance boost (**67.5%** on average). (2) $(\Omega, \theta, \Delta t)$ remains the most desired heterogeneity across datasets. (3) $(\mathbf{b}, \Omega, \theta, \Delta t)$ het-

*Table 5.* Accuracy v/s SOPs tradeoff. Additional learnable parameters $(u, v, b)$ originally set to zero in the HRF layer of $SH^2$RFSSM on EigenWorms (18k sequence). Energy comparison with LinOSS on the same hyperparameter combination. b learnable initalized heterogeneously (Boyer et al., 2025). Learnable $(\Omega, \theta, \Delta t)$ IMEX and IM models are further analysed in the main paper, Figure 5.

| Model | Learnable | | | | | | EigenWorms | | | |
|---|---|---|---|---|---|---|---|---|---|---|
| | **u** | **b** | $\Omega$ | $\theta$ | $\Delta t$ | **v** | **Accuracy** | **SOPs** ($\times 10^6$) | **Energy** | **Steps** ($\times 10^3$) |
| IMEX | ✗ | ✗ | ✓ | ✓ | ✓ | ✗ | 90.0±5.72 | 365.53±7.17 | 69.22× | 22.4± 4.4 |
| | ✗ | ✓ | ✓ | ✓ | ✓ | ✗ | 88.33±4.78 | 281.15±28.47 | 85.83× | 24.6± 7.1 |
| | ✓ | ✗ | ✓ | ✓ | ✓ | ✓ | 91.11±4.78 | 362.06±12.04 | 69.78× | 23.0± 4.5 |
| | ✓ | ✓ | ✓ | ✓ | ✓ | ✓ | 88.33±6.18 | 299.97± 27.77 | 81.47× | 23.4± 6.8 |
| IM | ✗ | ✗ | ✓ | ✓ | ✓ | ✗ | 91.11±3.68 | 403.51±14.92 | 63.68× | 17.6± 3.0 |
| | ✓ | ✗ | ✓ | ✓ | ✓ | ✓ | 91.11±4.78 | 395.17±12.63 | 64.82× | 17.4± 6.1 |

erogeneity offers **dataset-specific** boost in performance for SCP1. This selective performance improvement was also observed in (Boyer et al., 2025). (4) $(\mathbf{u}, \mathbf{\Omega}, \theta, \mathbf{\Delta t}, \mathbf{v})$ heterogeneity is observed to **reduce SOPs**; however, it is not necessarily always the optimal combination across datasets, as observed in both paradigms. (5) For $(\mathbf{\Omega}, \theta, \mathbf{\Delta t})$, **Encoder spike rates** (From Figure 5) remain **similar** across IM and IMEX discretisations, while **IM** shows **reduced spike activity** for $\theta_C$, consistent with trends in Table 1. Hence, application-specific selection of heterogeneity is important.

### 4.4.1. ENERGY ESTIMATION

To assess energy computation, similar hidden-size (H), state-size (P), and num-blocks (N) were used. We observe that our block performs no matrix multiplications and is well-suited for neuromorphic hardware. We estimate the theoretical energy consumption following prior works (Shen et al., 2025; Bal & Sengupta, 2025; Wu et al., 2025). Assuming 45nm hardware (Horowitz, 2014), a floating-point operation (FLOP) consumes $E_{\text{FLOPs}} = 4.6$pJ, while an accumulate (AC) operation consumes $E_{\text{AC}} = 0.9$pJ. A multiplication between a floating-point weight and a binary spike is treated as an addition-only operation. Hence, the energy consumption is computed as $4.6$pJ $\times$ FLOPs$(n)$ and $0.9$pJ $\times$ SOPs$(n)$ respectively. The synaptic operations (SOPs) are estimated as SOPs$(n) = f_r \times$ FLOPs$(n)$, where $f_r$ denotes the firing rate of the input spike train for the block/layer, FLOPs$(n)$ refers to the number of floating-point operations in layer $n$.

Since $\Omega$ is a time-invariant diagonal matrix, it can be efficiently implemented using parallel scans with computational time of order $\mathcal{O}(P \log(L))$ (Total computations are $\mathcal{O}(PL)$), while at inference can be precomputed. Also, D is $\mathcal{O}(HL)$, which is much smaller than $\mathcal{O}(LPH)$, i.e., computation for B & C matrices or even $\mathcal{O}(LH^2)$, which is for the GLU layer. Rusch & Rus (2025) uses GeLU and GLU non-linearities, which we replace with a linear layer. And, as described in Yu et al. (2023), GeLU consumes 14 FLOPs per operation. Moreover, GLU has twice as many FLOPs as a linear layer. Overall, we observe that, with the same hyperparameters and learnable parameters $(\Omega, \theta, \Delta t)$, the IM

model and IMEX model are $\sim$ **69×** more energy-efficient than the equivalent LinOSS model for the EigenWorms dataset (spike patterns observed in Figure 5).

### 4.5. Multivariate Long-term Forecasting

To evaluate the HRF neuron's signal processing, we propose SpikHRFSSM and test its generalisation on long-horizon multivariate forecasting benchmarks, including ETT, Electricity, Weather, Traffic, and Exchange. These datasets span diverse temporal resolutions and dimensionalities (refer to Table 14 in Appendix F). The architecture employs encoder-decoder framework similar to Wu et al. (2025).The proposed signal processing approach in Wu et al. (2025) is not optimal for neuromorphic hardware. Hence, we replace the non-learnable FFT and IFFT layers with our neurons implemented in an SSM framework as detailed above. For fair comparison, we reproduce our experiments using the best hyperparameter configurations reported by Wu et al. (2025) and evaluate performance using Mean Squared Error (MSE) and Mean Absolute Error (MAE) across all datasets. Following the baselines in Wu et al. (2025), we compare our model against transformer-based methods (Wu et al., 2021; Zhang & Yan, 2023; Nie et al., 2023; Liu et al., 2024), convolution-based approaches (Liu et al., 2022; Wu et al., 2023), and linear models (Zeng et al., 2023; Li et al., 2023). Full benchmark results across all methods are reported in Table 15, while Table 6 presents the average performance across different forecasting horizons for the top-performing models. For the ETT datasets, we additionally compare against frequency-domain methods (Xu et al., 2024; Zhou et al., 2022) and spiking neural network models (Wu et al., 2025; Lv et al., 2024; Kim et al., 2019). These baseline results are also adopted from Wu et al. (2025) for consistency and fair comparison, as summarised in Table 16. From Table 6, we observe strong performance results on the regression benchmark. SpikHRFSSM outperforms SpikF for both MSE and MAE on Weather (0.242 vs. 0.245 MSE), ETTh2 (0.368 vs. 0.372 MSE), ETTm2 (0.274 vs. 0.281 MSE), and Traffic (0.494 vs. 0.497 MSE). We observe a comparable performance to SpikF for ETTh1 and ETTm1. On ECL, we

*Table 6.* Best regression performance (mean for 3 runs) for multivariate time-series regression datasets. Top three models are highlighted in **red**, **blue**, **violet** respectively.

| Model | SpikHRFSSM (Ours) | | SpikF (2025) | | iTransformer (2024) | | RLinear (2023) | | PatchTST (2023) | | Crossformer (2023) | | TimesNet (2023) | | DLinear (2023) | |
|---|---|---|---|---|---|---|---|---|---|---|---|---|---|---|---|---|
| Metric | MSE | MAE | MSE | MAE | MSE | MAE | MSE | MAE | MSE | MAE | MSE | MAE | MSE | MAE | MSE | MAE |
| ECL | 0.195 | **0.275** | **0.183** | **0.275** | **0.178** | **0.27** | 0.219 | 0.298 | 0.205 | 0.29 | 0.244 | 0.334 | **0.192** | 0.295 | 0.212 | 0.3 |
| Weather | **0.242** | **0.264** | **0.245** | **0.265** | **0.258** | **0.278** | 0.272 | 0.291 | 0.259 | 0.281 | 0.259 | 0.315 | 0.259 | 0.287 | 0.265 | 0.317 |
| ETTh1 | **0.443** | **0.431** | **0.44** | **0.428** | 0.454 | 0.447 | **0.446** | **0.434** | 0.469 | 0.454 | 0.529 | 0.522 | 0.458 | 0.45 | 0.559 | 0.515 |
| ETTh2 | **0.368** | **0.393** | **0.372** | **0.394** | **0.383** | 0.407 | 0.414 | **0.398** | 0.387 | 0.407 | 0.942 | 0.684 | 0.414 | 0.427 | 0.954 | 0.723 |
| ETTm1 | **0.39** | **0.389** | **0.388** | **0.385** | 0.407 | 0.41 | 0.414 | 0.407 | **0.387** | **0.4** | 0.513 | 0.496 | 0.4 | 0.406 | 0.403 | 0.407 |
| ETTm2 | **0.274** | **0.316** | **0.281** | **0.32** | 0.288 | 0.332 | 0.286 | 0.327 | **0.281** | **0.326** | 0.757 | 0.61 | 0.291 | 0.333 | 0.35 | 0.401 |
| Traffic | **0.494** | **0.291** | 0.497 | **0.296** | **0.428** | **0.282** | 0.626 | 0.378 | **0.481** | 0.304 | 0.55 | 0.304 | 0.62 | 0.336 | 0.625 | 0.383 |
| Exchange | 0.388 | 0.417 | **0.36** | **0.402** | **0.36** | **0.403** | 0.378 | 0.417 | 0.367 | **0.404** | 0.94 | 0.707 | 0.416 | 0.443 | **0.354** | 0.414 |

observe our model ties with SpikF on MAE. SpikHRFSSM outperforms all other(spiking and ANN-based) benchmarks on the weather, ETTh2, and ETTm2 datasets. We observe that spike trains admit well-defined frequency-domain encodings through their responses to oscillatory kernels.

### 4.5.1. SIGNAL PROCESSING WITH AN HRF LAYER

To assess the filtering capabilities of the HRF neuron, we conducted experiments on the ETTh1 dataset with a prediction length of 720. We extracted spikes just before passing them through the HRF layer (Figure 6a), and compared the same slice with the one after passing through a layer of HRF neurons (Figure 6c). We observe that the HRF layer suppresses temporally isolated spikes (e.g., in patches 10 and 20), which likely correspond to noise-like events. As per heterogeneous frequencies (Figure 6b) and thresholds learned by our model, the post-HRF activity exhibits repeating spike patterns at regular temporal intervals (e.g., in patches 1 and 7). This suggests that HRF neurons exploit data-driven parameter heterogeneity to perform structured temporal filtering. Furthermore, because the IMEX discretisation preserves sustained oscillations in the undamped regime, HRF neurons can encode frequency-selective information through spike timing patterns.

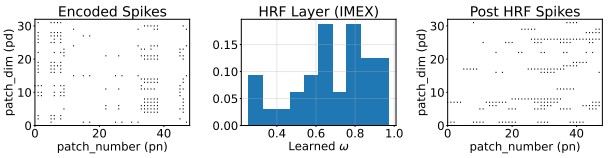

*Figure 6.* (a), (c) Pre-SSM and Post-neuron raster plots for a slice of the time series across patch dimensions and patch numbers.(b) different learned frequencies by the HRF layer, filtering out noise that does not match the relevant frequencies

### 4.5.2. COMPUTATIONAL EFFICIENCY COMPARISION

As mentioned previously, SpikHRFSSM adopts a neuromorphic hardware-friendly feature extraction, unlike SpikF (Wu et al., 2025). We further report the learnable parameters, peak memory, energy, and latency of our HRF layer with FFT-based Signal Processing (which comprises FFT, LIF projection, and an IFFT layer), as shown in Table 7. Since FFT and IFFT are non-learnable, the parameters account only for the LIF layer. From Table 7, for SpikHRFSSM, we observe a $2\times$ latency reduction and a $4\times$ theoretical energy efficiency as compared to SpikF (Wu et al., 2025) on the ETTh1 dataset. Moreover, we achieve this benefit with a comparable peak memory usage. The results show that HRF avoids complex operations and does not require time-to-frequency domain conversion, making RF neurons well suited for neuromorphic hardware implementation (Shrestha et al., 2024; Davies et al., 2021).

*Table 7.* Comparison between the proposed spiking HRF layer and frequency-based components used in SpikF (Wu et al., 2025).

| Layer | Params | Memory | Energy | Latency |
|---|---|---|---|---|
| FFT | 1.2k | 21.4 MB | 7.84 $\mu$J | 7.77 ms |
| **HRF** | 2.2k | 22.0 MB | **1.98 $\mu$J$_{\downarrow 4\times}$** | **3.93 ms$_{\downarrow 2\times}$** |

## 5. Discussion

We propose S$H^2$RFSSM, an energy-efficient second-order spiking SSM built with harmonic resonate-and-fire neurons and a learnable encoder and decoder. Unlike prior similar SSMs, S$H^2$RFSSM is fully spike-based, without GeLU/GLU, and tailored for very long-sequence modelling. The model performs well on HAR datasets while consuming significantly less energy, and also achieves superior classification and regression performance on 18k EigenWorms and 50k-length PPG datasets, respectively. Hence, it is ideally suited for resource-constrained AI applications such as healthcare wearable devices. Moreover, we conduct a detailed analysis of the impact of neuronal heterogeneities and discretisation methods on RF neurons. In addition, we propose SpikHRFSSM, which enables accurate and efficient spike signal processing for numerous multivariate long-range forecasting applications. Future work will focus on deploying this work for edge AI using Intel Loihi2 (Shrestha et al., 2024).

## Acknowledgment

KA, VN, and AB gratefully acknowledge the support provided by IIT Guwahati. Portions of this work were carried out by AV and VS during their summer internship under the supervision of AB. The authors also sincerely thank the anonymous reviewers for their insightful comments and constructive suggestions, which significantly improved the overall quality of this manuscript.

## Impact Statement

This work advances the field of machine learning. We believe this work offers numerous benefits across healthcare, finance, and other industry applications of AI. For instance, reducing AI energy consumption will lower its carbon footprint, thereby advancing Green AI. Moreover, the model's close proximity to biology also provides a framework for understanding brain computation.

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

# A. Harmonic-Resonate-and Fire Neurons

## A.1. From Resonate-and-Fire to Harmonic Resonate-and-Fire

The Resonate-and-Fire (RF) neuron (Izhikevich, 2001) extends the classical leaky integrate-and-fire model by incorporating intrinsic oscillatory dynamics, enabling neurons to selectively respond to specific input frequencies. In contrast to LIF neurons, RF neurons exhibit subthreshold resonance, a phenomenon observed in Hodgkin-Huxley-type neurons and in many biological oscillatory circuits.

The RF neuron is defined by a two-dimensional linear dynamical system

$$u'(t) = b\,u(t) - \omega\,v(t) + x(t), \tag{4}$$
$$v'(t) = \omega\,u(t) + b\,v(t), \tag{5}$$

where $u(t)$ and $v(t)$ represent the real and imaginary components of the complex membrane state, $x(t)$ is the input current, $\omega > 0$ is the intrinsic angular frequency, and $b < 0$ controls exponential damping. This system can be equivalently written in complex form as

$$w'(t) = (b + i\omega)\,w(t) + x(t), \qquad w(t) = u(t) + iv(t), \tag{6}$$

Following prior work (AlKhamissi et al., 2021; Higuchi et al., 2024), we adopt an equivalent real-valued reformulation of the RF dynamics, known as the Harmonic Resonate-and-Fire (HRF) model. This formulation rewrites the same oscillatory system as a second-order real dynamical system:

$$u'(t) = -\omega^2 v(t) - 2b\,u(t) + x(t), \tag{7}$$
$$v'(t) = u(t), \tag{8}$$

which avoids complex-valued state variables while preserving the underlying resonant behaviour.

Here, $v(t)$ corresponds to the membrane potential and $u(t)$ to its velocity. This formulation is equivalent to a damped harmonic oscillator driven by the input $x(t)$ and preserves the resonant behaviour of RF neurons while enabling stable and efficient real-valued computation. Importantly, prior work has shown that HRF neurons achieve their strongest memory and representational capacity when operated without membrane resets (Higuchi et al., 2024), making them particularly suitable for long-range temporal modelling.

HRF neurons are oscillating neurons that emulate phase properties similar to RF neurons, especially in the $b = 0$ regime with a time period of $2\pi/\omega$.

Consider the second-order HRF neuronal dynamics:

$$u'(t) = -\omega^2 v(t) - 2bu(t) + x(t),$$

$$v'(t) = u(t),$$

where $\omega > 0$, $b \in \mathbb{R}$, and $x(t)$ is an input. If we define the scaled coordinate $\tilde{v}(t) = \omega v(t)$, then the system can be equivalently written as

$$u'(t) = -\omega\tilde{v}(t) - 2bu(t) + x(t),$$

$$\tilde{v}'(t) = \omega u(t).$$

**Theorem A.1.** *(Scaled Equivalence of Undamped HRF neurons and Undamped RF neurons) For $b = 0$, the dynamics reduce to*

$$w'(t) = i\omega w(t) + x(t), \qquad w(t) = u(t) + i\tilde{v}(t).$$

*which coincide with the polar form of the complex-valued first-order system of an undamped RF neuron.*

*Proof.* Set $b = 0$ in the scaled system:

$$u'(t) = -\omega\tilde{v}(t) + x(t),$$

$$\tilde{v}'(t) = \omega u(t).$$

Define the complex variable $w(t) = u(t) + i\tilde{v}(t)$. Then

$$w'(t) = u'(t) + i\tilde{v}'(t)$$
$$= \left(-\omega\tilde{v}(t) + x(t)\right) + i(\omega u(t))$$
$$= i\omega(u(t) + i\tilde{v}(t)) + x(t)$$
$$= i\omega w(t) + x(t).$$

which matches the stated polar dynamics. □

*Remark* A.2. The scaling $\tilde{v} = \omega v$ removes the anisotropy induced by $\omega$ in the original $(u, v)$ coordinates. Thus, the apparent discrepancy between the second-order real formulation and the first-order complex formulation arises purely from coordinate scaling.

**Spiking mechanism** Spikes are generated by thresholding the internal state,

$$z(t) = \Theta(u(t) - \theta), \tag{9}$$

where $\Theta$ denotes the Heaviside step function and $\theta$ is a learnable threshold.

## B. Forced Damped Harmonic Oscillator as a Temporal Kernel

We consider the forced second-order linear system

$$v''(t) + 2b\,v'(t) + \omega^2 v(t) = I(t), \tag{10}$$

where $b \geq 0$, $\omega > 0$, and $I(t)$ is an external input. We focus on the underdamped regime

$$\omega_d := \sqrt{\omega^2 - b^2} > 0,$$

and assume zero initial conditions.

**Closed-Form Solution:** The unique causal solution of (10) can be written as

$$v(t) = \int_0^t I(\tau)\,k(t - \tau)\,d\tau, \tag{11}$$

where the kernel

$$k(s) = \frac{1}{\omega_d}\,e^{-bs}\,\sin(\omega_d s), \qquad s \geq 0, \tag{12}$$

acts as a causal, exponentially windowed sinusoid.

**Verification:** Differentiating (11) using Leibniz' rule yields

$$v'(t) = \int_0^t I(\tau)\,k'(t - \tau)\,d\tau,$$

since $k(0) = 0$. Differentiating once more gives

$$v''(t) = \int_0^t I(\tau)\,k''(t - \tau)\,d\tau + I(t), \tag{13}$$

where $k'(0) = 1$.

A direct calculation shows that the kernel satisfies

$$k''(s) + 2b\,k'(s) + \omega^2 k(s) = 0. \tag{14}$$

Substituting (14) into (13) yields

$$v''(t) + 2b\,v'(t) + \omega^2 v(t) = I(t),$$

confirming that (11) solves (10).

**Impulse Response Interpretation:** Since the system is linear and time-invariant, the kernel $k(s)$ is precisely the impulse response. Setting $I(t) = \delta(t)$ in (11) yields

$$v(t) = k(t),$$

recovering (12) as the causal Green's function of the damped oscillator.

**Undamped Limit and Phase Structure** In the limit $b \to 0$, we obtain

$$k(s) \to \frac{1}{\omega}\sin(\omega s),$$

and the solution reduces to

$$v(t) = \int_0^t I(\tau) \frac{1}{\omega} \sin\big(\omega(t-\tau)\big)\, d\tau, \tag{15}$$

corresponding to a causal time–windowed sine transform with infinite memory and marginal stability.

Differentiating (15) gives

$$v'(t) = \int_0^t I(\tau) \cos\big(\omega(t-\tau)\big)\, d\tau = \omega \int_0^t I(\tau) \frac{1}{\omega}\sin\big(\omega(t-\tau) + \tfrac{\pi}{2}\big)\, d\tau. \tag{16}$$

Thus, in the undamped regime, $v(t)$ and $v'(t)$ correspond to the **same oscillatory temporal kernel** with a **fixed phase shift** of 90° and a scaling by $\omega$. Thresholding either variable therefore produces spikes driven by the same filtered signal, differing only in timing within each oscillation cycle. These differences do not alter representational capacity and can be absorbed into downstream linear readout weights. This observation justifies the use of either state variable for spike generation in undamped HRF neurons.

**Amplitude–Phase Structure in the Damped Case** For $b > 0$, the kernel derivative

$$k'(s) = e^{-bs}\left[\cos(\omega_d s) - \frac{b}{\omega_d}\sin(\omega_d s)\right] \tag{17}$$

is a linear combination of sine and cosine. Using

$$A\cos x + B\sin x = \sqrt{A^2 + B^2}\,\sin(x + \phi), \qquad \phi = \arctan\left(\frac{A}{B}\right),$$

We can rewrite (17) as a single exponentially decaying sinusoid with a phase shift. Unlike the undamped case, where differentiation induces a pure 90° phase rotation, damping couples amplitude decay and phase evolution, breaking the simple rotational structure between $v(t)$ and $v'(t)$.

The forced damped oscillator, therefore, computes a causal, frequency-selective temporal projection of the input signal. The exponential envelope controls memory and stability, while the oscillatory component preserves resonance, providing a compact dynamical mechanism for temporal filtering.

## C. Discretisation and Neuron Heterogeneity

### C.1. Discretisation of Oscillatory Spiking Dynamics

Unlike LIF neurons, oscillatory neurons such as RF and HRF are fundamentally sensitive to time discretisation. Because their dynamics encode energy-preserving phase-space rotations, the numerical integrator becomes part of the model itself: inappropriate discretisation can destroy oscillations, introduce artificial damping, or lead to instability.

In continuous time, HRF dynamics are stable for $b \geq 0$ and exhibit sustained oscillations when $b = 0$. However, standard explicit discretisations, such as the Euler forward method, do not preserve these properties and can introduce spurious energy growth, leading to divergence even for stable continuous systems. This makes the choice of numerical scheme critical when deploying oscillatory neurons in long-sequence learning. Discrete-time update rules for the harmonic resonate-and-fire neuron Let $\Delta t$ be the timestep, and subscript $n$ denote the $n^{th}$ timestep.

**Explicit Euler (EX):**   computationally cheap but unstable for oscillatory systems.

$$u_n = u_{n-1} + \Delta t \cdot (-2b\,u_{n-1} - \omega_0^2 v_{n-1} + I_n) \tag{18}$$
$$v_n = v_{n-1} + \Delta t \cdot u_{n-1} \tag{19}$$

**Implicit Euler (IM):**   unconditionally stable but introduces numerical dissipation that damps oscillations over time.

$$u_n = u_{n-1} + \Delta t \cdot (-2b\,u_n - \omega_0^2 v_n + I_n) \tag{20}$$
$$v_n = v_{n-1} + \Delta t \cdot u_n \tag{21}$$

**Implicit–Explicit (IMEX):**   treats the stiff oscillatory terms implicitly and the input terms explicitly, preserving stability while maintaining oscillatory energy. We can treat the damping b both implicitly, i.e., IMEX(IM b), or explicitly, i.e., IMEX(EX b).

$$u_n = u_{n-1} + \Delta t \cdot (-2b\,u_{\bar{n}} - \omega_0^2 v_{n-1} + I_n) \tag{22}$$
$$v_n = v_{n-1} + \Delta t \cdot u_n \tag{23}$$
$$where,\, u_{\bar{n}} = u_n, u_{n-1} \tag{24}$$

### C.2. Heterogeneity in HRF neurons

To address continuous spiking and instability observed in the basic RF neuron, we incorporate adaptive thresholding and a smooth reset mechanism, as proposed in the balanced HRF (BHRF) formulation (Higuchi et al., 2024). In addition, we introduce heterogeneity directly into the oscillator parameters, enabling a richer spectrum of temporal dynamics.

**State and Parameter Heterogeneity (AlKhamissi et al., 2021):**   The states $(u_n, v_n)$ form a damped harmonic oscillator in phase space. Neuronal diversity is introduced through heterogeneous intrinsic frequency $\omega$ and damping $b$. Larger $\omega$ yields faster oscillations and sensitivity to rapid input changes, while smaller $\omega$ supports slower temporal integration. Smaller $b$ leads to sustained oscillatory memory, whereas larger $b$ produces quickly damped responses. Additional diversity arises from different initial states and spike-triggered modulation of $(u_n, v_n)$.

**Adaptive Threshold (Higuchi et al., 2024):**   Spiking sparsity is induced by introducing a refractory variable $q_n$ into the threshold:

$$z_n = \Theta(u_n - \theta - q_n), \tag{25}$$
$$q_n = \gamma q_{n-1} + z_{n-1}, \tag{26}$$

where $\theta$ is a constant baseline threshold and $\gamma \in (0, 1)$ controls the exponential decay of the refractory effect. This adaptive threshold increases immediately after a spike and gradually relaxes, suppressing repeated firing.

**Soft Reset Mechanism (Higuchi et al., 2024):**   A further limitation of the basic RF neuron is the conventional hard reset, which abruptly reduces the state amplitude and perturbs the intrinsic oscillatory dynamics. To address this, the BHRF model replaces the hard reset with a smooth reset by integrating the refractory variable into the damping term (Higuchi et al., 2024):

$$b_n = b + q_n, \tag{27}$$

After a spike, the effective damping is temporarily increased, allowing the oscillation amplitude to decay more rapidly while preserving phase continuity, followed by a soft reset of the internal states. The resulting soft reset of the state variables is expressed as

$$u_n = u_{n-1}(1 - z_n\theta\zeta), \tag{28}$$
$$v_n = v_{n-1}(1 - z_n\theta\zeta), \tag{29}$$

with $\zeta$ controlling the strength of the reset.

**Special Case: Fixed Threshold (HRF).**  When the refractory dynamics are disabled by setting

$$q_n = 0 \quad \forall n, \tag{30}$$

The threshold becomes time-invariant. In this case, the model reduces to the standard harmonic resonate-and-fire (HRF) neuron, recovering its original spiking behaviour without adaptive thresholding or refractory-induced sparsity.

### C.3. Optimal Heterogeneity Search on Spiking Heidelberg Dataset (SHD)

In this section, we examine how introducing heterogeneity can enhance a neuron's learning capabilities. We conducted systematic experiments with different initialisations and grid search on neuronal parameter settings $(u, b, \omega, \theta, \Delta t, v)$, along with discretisations, to achieve good performance for HRF and BHRF neurons. We start with a BHRF neuron with a Forward Euler Discretisation (Higuchi et al., 2024). We initialise all parameters to a single value and set all model parameter combinations learnable. We report Accuracy and Spiking Operations (SOPs) for all combinations (see Figure 7). Since we observed that learnable $\Delta t$ does not improve performance and instead degrades it, we set $zeta = 0.5$ and $\Delta t = 0.01$. We later heterogeneously initialise our neurons as per Higuchi et al. (2024) using the best-performing learnable parameter combinations to boost performance. We also explore learnable $\zeta$ and other discretisations in Table 1.

## D. SSM Formulation

To study the theoretical properties of a second-order ODE in $\mathrm{S}H^2\mathrm{RFSSM}$ and SpikHRFSSM, we draw upon the studies in Rusch & Rus (2025). Our model, like theirs, can be formulated as energy-conserving and dissipative.

- **IM discretization (Rusch & Rus, 2025):** We consider the implicit (backward Euler) discretisation of a second-order system involving a position-like state $v_n$ and a velocity-like state $u_n$, similar to Rusch & Rus (2025). The implicit scheme is known to introduce additional dissipative terms that enhance the dynamics' stability, particularly in the presence of stiffness.

  The discretised updates are given by:

  $$u_n = u_{n-1} + \Delta t \left(-\Omega v_n + B x_n\right),$$
  $$v_n = v_{n-1} + \Delta t \, u_n,$$

  where $\Omega$ is a diagonal matrix of oscillation frequencies and $B$ is an input projection matrix. Note that both $u_n$ and $v_n$ are evaluated at the future timestep, in contrast to explicit methods.

  Letting the concatenated state be $s_n$, the above system can be written compactly as:

  $$M s_n = s_{n-1} + F_n,$$

  where

  $$M = \begin{pmatrix} I & \Delta t \Omega \\ -\Delta t I & I \end{pmatrix}, \qquad F_n = \begin{pmatrix} \Delta t B x_n \\ 0 \end{pmatrix}.$$

  To obtain an explicit update rule, we algebraically solve the coupled system by introducing the matrix inverse $S = (I + \Delta t^2 \Omega)^{-1}$. Substituting and simplifying yields:

  $$s_n = M^{\mathrm{IM}} s_{n-1} + F_n^{\mathrm{IM}}, \tag{31}$$

  where

  $$M^{\mathrm{IM}} = \begin{pmatrix} S & -S \Delta t \Omega \\ S \Delta t & S \end{pmatrix}, \qquad F_n^{\mathrm{IM}} = \begin{pmatrix} S \Delta t B x_n \\ S \Delta t^2 B x_n \end{pmatrix}.$$

  This formulation highlights the stabilising effect of the implicit method: the matrix $S = (I + \Delta t^2 \Omega)^{-1}$ is a Schur complement that acts as a preconditioner, suppressing high-frequency components. Consequently, the eigenvalues of $M^{\mathrm{IM}}$ remain bounded within the unit circle for a wide range of $\Delta t$, leading to improved numerical stability. The Schur complement can be computed in $\mathcal{O}(m)$ instead of the typical $\mathcal{O}(m^3)$ operations using Gauss-Jordan elimination(Rusch & Rus, 2025).

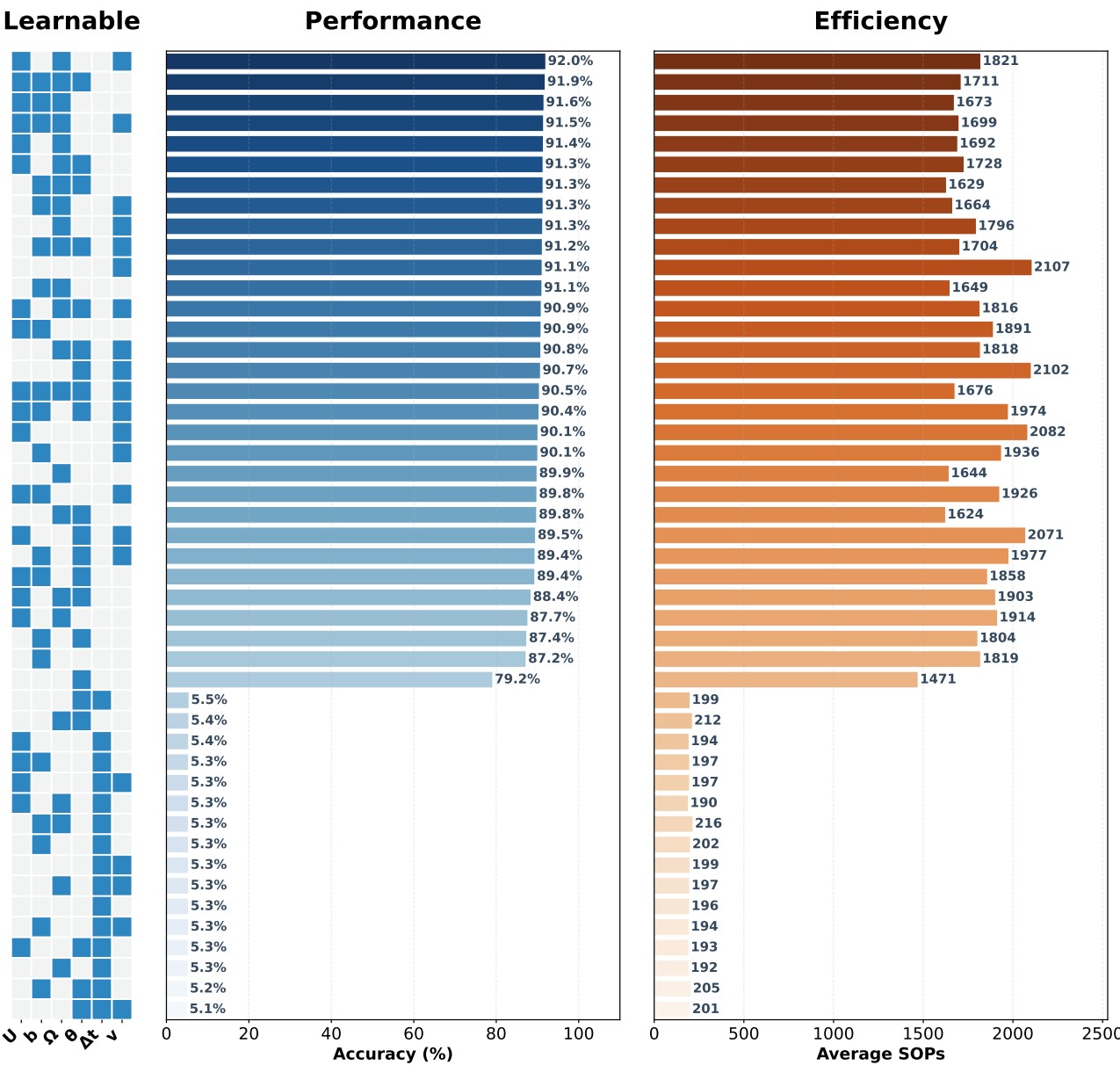

*Figure 7.* Model Performance on Spiking Heidelberg Dataset by setting learnable neuron parameters on homogeneously initialised HRF neuron

- **IMEX discretization (Rusch & Rus, 2025):** We also utilise an implicit-explicit (IMEX) scheme for discretising the second-order harmonic oscillator system, similar to Rusch & Rus (2025). IMEX methods treat the stiff terms implicitly and the non-stiff or input terms explicitly, resulting in a balanced scheme that enables stable yet undamped oscillations. As shown in Rusch & Rus (2025), such schemes preserve the total energy of the system and therefore are particularly well-suited for learning long-range sequential patterns without introducing artificial dissipation.

The update equations under the IMEX discretisation are given by:

$$u_n = u_{n-1} + \Delta t \left(-\Omega v_{n-1} + B x_n\right),$$
$$v_n = v_{n-1} + \Delta t \, u_n,$$

where the velocity update depends implicitly on the newly computed $u_n$, while the force term $-\Omega v_{n-1} + B x_n$ is evaluated using previous state values.

Defining the state vector as $s_n$, we can rewrite the update in matrix form:

$$M s_n = M_1 s_{n-1} + F_n,$$

where the matrices $M$, $M_1$, and input vector $F_n$ are:

$$M = \begin{pmatrix} I & 0 \\ -\Delta t I & I \end{pmatrix}, M_1 = \begin{pmatrix} I & -\Delta t \Omega \\ 0 & I \end{pmatrix}, F_n = \begin{pmatrix} \Delta t B x_n \\ 0 \end{pmatrix}.$$

Multiplying both sides by $M^{-1}$ yields the closed-form update:

$$s_n = M^{\text{IMEX}} s_{n-1} + F_n^{\text{IMEX}}, \tag{32}$$

where the transition matrix and input vector are given by:

$$M^{\text{IMEX}} = \begin{pmatrix} I & -\Delta t \Omega \\ \Delta t I & I - \Delta t^2 \Omega \end{pmatrix}, \qquad F_n^{\text{IMEX}} = \begin{pmatrix} \Delta t B x_n \\ \Delta t^2 B x_n \end{pmatrix}.$$

### D.1. Phase Space Properties of the HRF neuron

We can emulate RF neurons with real-valued states and second-order dynamics given the equivalence(Theorem A.1). From the phase diagram (Figure 8ss), we observe elliptical orbits due to the $\omega$ scaling. Such dynamics upon discretisation using IMEX or IM exhibit stability. IMEX discretisation also preserves oscillations, although they follow a different trajectory than in the continuous-time formulation. However, IM dynamics spiral down to the fixed point.

In typical RNNs and SSMs, past information is encoded in hidden states that decay exponentially over time. In contrast, sustained oscillatory dynamics encode information in the state's phase and amplitude, preventing decay. Intuitively, this corresponds to information rotating in state space, allowing it to be preserved over long horizons.

### D.2. Parallel scan (Rusch & Rus, 2025)

Parallel scans (Kogge & Stone, 1973; Blelloch, 1990) exploit associativity to reduce recurrent computation from $\mathcal{O}(N)$ to $\mathcal{O}(\log N)$. Originally developed for RNNs, they have recently been adapted to state-space models (Smith et al., 2023), enabling efficient architectures such as LRUs (Orvieto et al., 2023) and Mamba (Gu & Dao, 2024). In our setting, parallel scans accelerate linear updates, with spike functions applied afterwards. We utilise an associative binary operation similar to Rusch & Rus (2025):

$$(a_1, a_2) \bullet (b_1, b_2) = (b_1 \cdot a_1, \; b_1 \cdot a_2 + b_2), \tag{33}$$

where $\cdot$ denotes matrix-matrix or matrix-vector multiplication. Applying a parallel scan to the input sequence $\{(M, F_n)\}$ efficiently solves

$$s_n = M s_{n-1} + F_n, \tag{34}$$

with the second tuple element storing $x_n$. Efficiency is achieved by exploiting structured matrices (e.g., diagonal block $2 \times 2$ forms in $M_{IM}$ and $M_{IMEX}$), where each multiplication is linear in the hidden dimension. We use this formulation to implement IM and IMEX discretisations for HRF neurons.

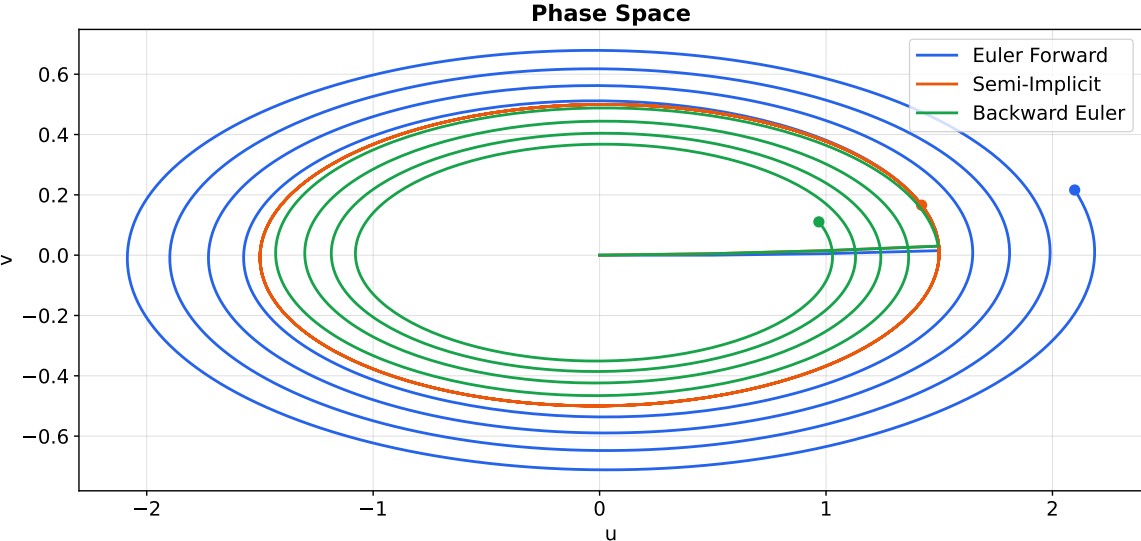

*Figure 8.* Phase Space orbits of HRF Neuron (damp set to 0) under different discretization schemes. The continuous system exhibits stable elliptical orbits that preserve information via phase rotation. The IMEX scheme maintains these long-horizon oscillations. In contrast, the IM scheme spirals inward to the fixed point due to over-damping, while the EX scheme spirals outward due to numerical instability.

### D.3. Dataset Description for S$H^2$RFSSM

**UEA Benchmark:** We evaluate on six long-sequence datasets from the University of East Anglia Multivariate Time Series Classification Archive (UEA-MTSCA) (Bagnall et al., 2018), which consist of real-world multivariate time-series classification tasks.

**PPG:** To assess performance on tasks with very long temporal dependencies, we consider the PPG-DaLiA dataset for regression, comprising 49,920 sequences. PPG-DaLiA involves heart rate estimation from wrist-worn sensor signals collected from 15 participants over 150 minutes at a sampling rate of 128 Hz.

**HAR:** We further evaluate our model on human activity recognition (HAR) benchmarks, namely UCI-HAR (Anguita et al., 2013) and UniMiB-SHAR (Micucci et al., 2017). The UCI-HAR dataset contains 10.3k samples from 30 subjects performing six activities (walking, walking upstairs, walking downstairs, sitting, standing, and lying), recorded using a tri-axial accelerometer and gyroscope at 50 Hz on a Samsung Galaxy SII smartphone. The UniMiB-SHAR dataset includes 11.7k samples spanning 17 activities (nine daily activities and six fall types), captured with a tri-axial accelerometer at up to 50 Hz using a Samsung Galaxy Nexus I9250 device.

### D.4. Why existing spiking SSMs fail on very long sequences?

State-of-the-art Spiking SSMs Bal & Sengupta (2025); Shen et al. (2025) augment Mamba-style SSMs with additional LIF neuron states for spiking memory. This enables them to perform better. However, it inherits Mamba's performance limitations and fails to outperform it. RF-based methods have been shown to outperform them on LRA benchmarks Zhang et al. (2026). However, they also fall short of the S4 baseline. We propose incorporating biologically plausible $\omega$-scaled undamped RF neuronal states, discretised using IM and IMEX, to capture stable ultra-long-range dependencies. In this undamped setting, the IMEX eigenvalues lie on the unit circle, thereby avoiding decay and enabling stable learning of long-range dependencies. For IM discretization, even under the largest observed settings ($\Delta t_{\max} = 0.826$, $\Omega_{\max} = 0.992$, $N = 17984$ for the EigenWorms dataset; see Figure 9), the expected magnitude of the $N$-th power of the eigenvalues remains on the order of $10^{-4}$ ($\approx 1.64 \times 10^{-4}$) Rusch & Rus (2025), indicating that the signal does not vanish excessively even for long sequences directly benefiting the gradient flow through the spike functions adjusting the firing threshold.

## D.5. Extended Studies

### D.5.1. DATASET DISTRIBUTION

We analyse the learned parameter distributions for the best models across discretisations and heterogeneities (Table 4) to understand how the model adapts its dynamics across datasets. The input mixing vector (D) shows the highest variability, indicating its central role in adapting spike injection to dataset-specific temporal structure. In contrast, B and C remain concentrated, suggesting that the spike transformation mechanism is largely stable across tasks.

The threshold ($\theta$) stays within ($[0,1]$) but exhibits higher variance for EthanolConcentration, implying that heterogeneous firing thresholds improve robustness under complex or noisy dynamics. The time-step ($\Delta t$) consistently remains close to 1, enabling stable gradients—an effect supported by IM and IMEX discretisations.

Stability is further governed by the interplay between $\Omega$ and b, where ($b < \Omega$). In general, setting b to zero ensures stable, long-range learning. Interestingly, SelfRegulationSCP1 can learn stable interactions by introducing an additional b learnable parameter (Similar trends were observed in (Boyer et al., 2025)). Finally, Batch Normalisation improves performance for SCP1 and MotorImagery while effectively encoding data-specific patterns.

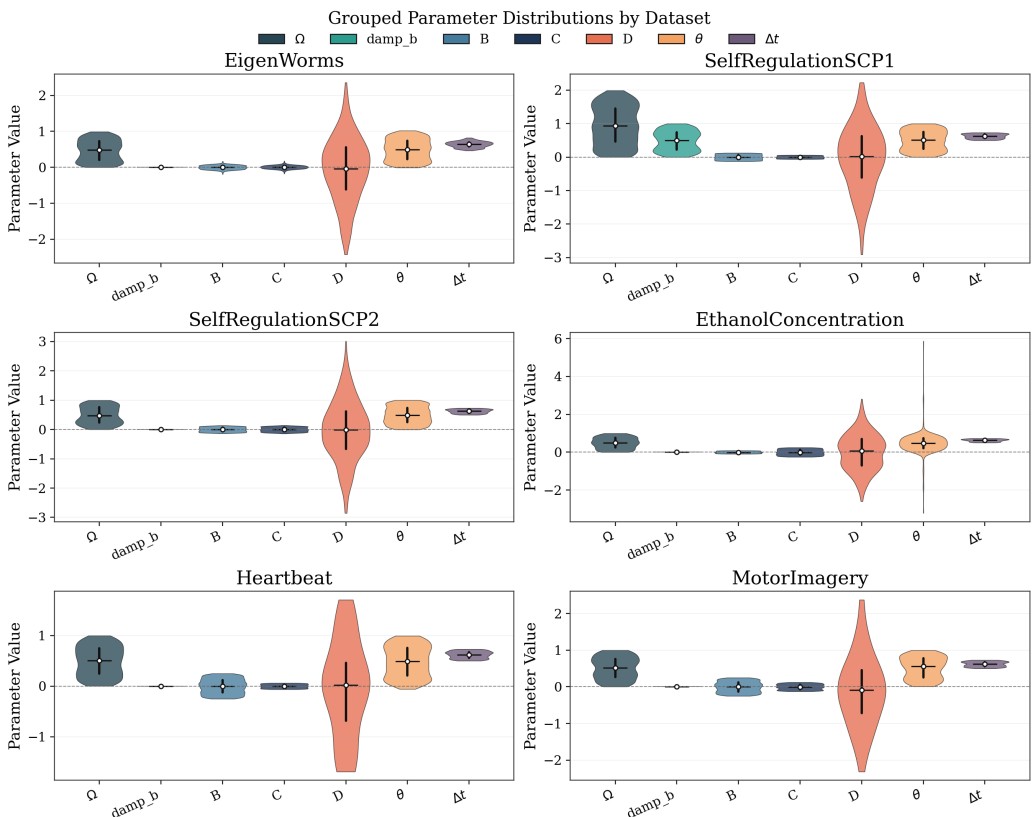

*Figure 9.* Ablation on the effect of kernel size on MSE for different training steps

### D.5.2. NORMALISATION

Detailed heterogeneity analysis on the selected UEA benchmark datasets is reported in Table 8. We compare different discretisation schemes and the effect of batch normalisation (BN) against prior oscillator-based baselines evaluated under a similar setup. Overall, the proposed S$H^2$RFSSM variants achieve competitive performance while additionally exhibiting spiking neural dynamics across architectures. Among the non-spiking baselines, LinOSS (IM) achieves the best average accuracy of 67.8%, outperforming the IMEX variant (65.0%), indicating the advantage of implicit discretisation for stable long-sequence classification modelling. Similarly, within the proposed models, S$H^2$RFSSM (IM) consistently outperforms S$H^2$RFSSM (IMEX), both with and without BN. The best-performing spiking configuration, S$H^2$RFSSM (IM) w/o BN, achieves an average accuracy of 66.5%, remaining within 1.3% of the best non-spiking baseline while maintaining

event-driven spiking behaviour.

We observe that removing batch normalisation improves performance across $SH^2$RFSSM variants, with the IM configuration improving from 64.4% to 66.5% average accuracy and the IMEX configuration improving from 64.2% to 65.3%. This suggests that normalisation may suppress amplitude-dependent oscillatory dynamics important for resonate-and-fire behaviour. The heterogeneous IMEX variant with learnable implicit damping parameter $b$ achieves competitive performance on SCP2 (86.8%), but reduces the overall average to 63.5%, indicating that excessive heterogeneity may introduce optimisation instability on certain datasets. Overall, these results demonstrate that second-order heterogeneous spiking oscillator dynamics can achieve performance comparable to strong linear oscillator baselines while preserving biologically-inspired sparse spiking computation.

*Table 8.* Mean and Standard Deviation reported for second-order methods on UEA dataset. $SH^2$RFSSM variants-w/ and w/o BN

| Seq. Length/ Class | | 17,984/5 | 896/2 | 1,152/2 | 1,751/4 | 405/2 | 3,000/2 | |
|---|---|---|---|---|---|---|---|---|
| | **SNN** | **Worms** | **SCP1** | **SCP2** | **Ethanol** | **Heartbeat** | **Motor** | **Avg** |
| LinOSS (IM) | No | $95.0 \pm 4.4$ | $87.8 \pm 2.6$ | $58.2 \pm 6.9$ | $29.9 \pm 0.6$ | $75.8 \pm 3.7$ | $60.0 \pm 7.5$ | 67.8 |
| LinOSS (IMEX) | No | $80.0 \pm 2.7$ | $87.5 \pm 4.0$ | $58.9 \pm 8.1$ | $29.9 \pm 1.0$ | $75.5 \pm 4.3$ | $57.9 \pm 5.3$ | 65.0 |
| BioOSS* | No | $92.8 \pm 5.2$ | $85.6 \pm 3.9$ | $55.1 \pm 1.8$ | $33.4 \pm 10.7$ | $74.8 \pm 2.0$ | $55.8 \pm 5.8$ | 66.3 |
| Lin-BPTT | No | $78.3 \pm 7.5$ | $86.4 \pm 1.8$ | $63.9 \pm 7.3$ | $29.9 \pm 0.6$ | $73.9 \pm 0.6$ | $48.1 \pm 5.7$ | 63.4 |
| Lin-RHEL | No | $75.0 \pm 9.9$ | $86.1 \pm 2.9$ | $61.4 \pm 9.4$ | $29.9 \pm 0.6$ | $73.5 \pm 1.6$ | $51.6 \pm 5.0$ | 62.9 |
| Nonlin-BPTT | No | $51.1 \pm 7.2$ | $86.8 \pm 3.2$ | $54.0 \pm 4.9$ | $29.9 \pm 0.6$ | $74.5 \pm 2.4$ | $56.5 \pm 7.6$ | 58.8 |
| Lin-RHEL | No | $50.6 \pm 6.7$ | $85.6 \pm 4.4$ | $54.0 \pm 2.0$ | $29.9 \pm 0.6$ | $73.9 \pm 4.3$ | $53.0 \pm 5.7$ | 57.8 |
| $SH^2$RFSSM (IM) w/ BN | Yes | $85.6 \pm 8.1$ | $82.6 \pm 4.5$ | $55.4 \pm 6.9$ | $30.1 \pm 1.47$ | $74.1 \pm 5.9$ | $\mathbf{59.6 \pm 5.4}$ | 64.4 |
| $SH^2$RFSSM (IMEX) w/ BN | Yes | $85.6 \pm 2.7$ | $82.6 \pm 4.2$ | $53.0 \pm 4.8$ | $32.4 \pm 4.6$ | $73.2 \pm 5.73$ | $58.6 \pm 7.3$ | 64.2 |
| $SH^2$RFSSM (IMEX w/ IM b) w/ BN | Yes | $83.3 \pm 4.6$ | $\mathbf{86.8 \pm 2.0}$ | $57.2 \pm 5.2$ | $26.3 \pm 7.0$ | $69.4 \pm 3.8$ | $51.2 \pm 4.4$ | 63.5 |
| $SH^2$RFSSM (IM) w/o BN | Yes | $\mathbf{92.8 \pm 3.3}$ | $84.2 \pm 3.2$ | $\mathbf{59.3 \pm 7.7}$ | $30.1 \pm 1.47$ | $\mathbf{74.5 \pm 3.4}$ | $58.2 \pm 3.2$ | 66.5 |
| $SH^2$RFSSM (IMEX) w/o BN | Yes | $90.0 \pm 5.7$ | $82.6 \pm 4.2$ | $53.0 \pm 4.8$ | $\mathbf{34.7 \pm 7.1}$ | $73.2 \pm 5.73$ | $58.6 \pm 7.3$ | 65.3 |

We conducted normalisation ablations on the encoder (Table 9) and highlight the following observations. We observe a clear dependence on sequence length. For shorter sequences, BN provides modest gains, likely by stabilising activation scale early in training. In contrast, for longer sequences, removing normalisation consistently improves both accuracy and variance. We hypothesise that, in this regime, the structured state-space dynamics, together with heterogeneous per-channel thresholds, are sufficient to regulate signal scale over time, rendering explicit normalisation unnecessary. Moreover, BN/LN introduce input-dependent rescaling of pre-threshold activations, which can interfere with the learned thresholds and disrupt temporally consistent spike generation.

*Table 9.* Effect of normalisation in the spike encoder. Here, LN refers to the Layer Normalisation and BN refers to the Batch Normalisation introduced in the Spike Encoder and post-SSM Linear layer, just before the spike function.

| Spike Encoder | Worms | SCP2 | Heartbeat |
|---|---|---|---|
| w/ LN | $79.4 \pm 17.3$ | $51.9 \pm 6.9$ | $73.2 \pm 2.2$ |
| w/ BN | $85.6 \pm 8.1$ | $55.4 \pm 6.9$ | $74.1 \pm 5.9$ |
| w/o Norm | $92.8 \pm 3.3$ | $59.3 \pm 7.7$ | $74.5 \pm 3.0$ |

**D.6. Role of Heterogeneous Initialisation**

Given the established sensitivity of SSMs to initializations, $SH^2$RFSSM is incorporated with heterogeneities for components - $\Omega$, B, C, D matrices and $\Delta t$, of the $SH^2$RFSSM layer, and the thresholds for the C ($\theta_C$), D ($\theta_D$) matrices and, the encoder ($\theta_{\text{Enc}}$). We did a detailed heterogeneity study on our IMEX model.

We studied the effect of each component's heterogeneity on performance by making them homogeneous (initialised by a constant value across all neurons). We start our experiments on IMEX with a Batch Normalisation Layer in our architecture (Table 4). From Table 10, we observe that homogeneous initialisation for neuronal thresholds leads to decreased performance. For homogeneous, big-step-size, we see a performance drop of 2.8% from 85.6 (See Table 10). Further, it drops by 3.4% and 4.5% when the $\Omega$ and D matrices are initialised with frequencies of 1 and 0, respectively, indicating the importance of heterogeneous states in these networks. It drops by 5% if D is initialised with zero. This implies, it is desirable to mix the HRF output with the input spikes. Homogeneity in thresholds in the Encoder layer doesn't reduce performance much (only 0.6%), but for thresholds C and D it drops by 1.7%, suggesting the importance of heterogeneous initialisation for

*Table 10.* Studying the role of homogeneously initialised components in an HRF Block on the EigenWorms dataset for five seeds on the best IMEX model(Here, $x^* = x^{-1/2}$).

| Parameters | Heterogeneity | Set Value | Accuracy |
|---|---|---|---|
| $\theta_{\text{Enc}}$ | $\mathcal{U}(0,1]$ | 1.0 | $85.0 \pm 7.2$ |
| $\Omega$ | $\mathcal{U}(0,1]$ | 1.0 | $82.2 \pm 3.3$ |
| B | $\mathcal{U}(-H^*, H^*)$ | 0.0 | $85.0 \pm 4.5$ |
| C | $\mathcal{U}(-P^*, P^*)$ | 0.0 | $85.0 \pm 4.5$ |
| D | $\mathcal{N}(0,1)$ | 0.0 | $80.5 \pm 4.6$ |
| $\Delta t$ | $\mathcal{U}(0,1]$ | 1.0 | $82.8 \pm 2.7$ |
| $\theta_C$ | $\mathcal{U}(0,1]$ | 1.0 | $83.3 \pm 3.9$ |
| $\theta_D$ | $\mathcal{U}(0,1]$ | 1.0 | $85.0 \pm 4.8$ |
| $\Omega$,B,C,D | As Above | As Above | $85.5 \pm 4.1$ |
| $\theta_C, \theta_D$ | As Above | As Above | $83.9 \pm 5.1$ |
| All | As Above | As Above | $85.5 \pm 4.1$ |

HRF neurons. We also noticed that complete homogenization of the model didn't significantly change the mean accuracy; rather, the deviation nearly doubled. Hence, we conclude that heterogeneous initialisation steers optimal model performance alongside heterogeneous learning.

### D.7. Ablation on Model Components

We further conducted ablations on model components for the above model. Interestingly, if we just consider the SSM layer in an S$H^2$RFSSM block, we observe a boost in model performance for EigenWorms by $1.6\%$, possibly because of higher representation power without the linear layer and subsequent threshold ($\theta_{Linear}$). Further, removing Batch Normalisation for EigenWorms in IMEX discretisation results in a 5% improvement over Batch Normalisation, with a performance of $90.0 \pm 5.7$. We observe similar trends for other datasets in Table 4. Further ablations on different normalisations are provided in the appendix 9. Nonlinearities such as GeLU yield a further 2.2% improvement, bringing the score to $92.2 \pm 5.1$. However, we don't include those in our architecture to account for a purely spike-based neuromorphic hardware optimal implementation.

*Table 11.* Ablation of S$H^2$RFSSM-IMEX components on the EigenWorms dataset. Although the best results use GeLU/GLU non-linearities in the SSM block, we adopt a fully spike-based variant (S$H^2$RFSSM without batch normalisation) as the baseline.

| Model Components | Accuracy |
|---|---|
| w/ BN | $85.6 \pm 2.7$ |
| w/ BN (Only SSM in Block) | $87.2 \pm 3.3$ |
| w/o BN | $90.0 \pm 5.7$ |
| w/o BN w/ GeLU | $92.2 \pm 5.1$ |

### D.8. Linear Layers in SSM Blocks

While performing ablation on the EigenWorms dataset, we observed that keeping only SSM in the S$H^2$RFSSM block improved performance by $1.6\%$ (Table 11). We extended our ablation to the PPG dataset to check if it can help in improving performance (Table 12). We observe that IM discretisation without a linear layer improves the MSE by $0.008$ with the same hyperparameters. But for IMEX, the MSE is better when we include the linear layer(by $0.006$). Hence, we observe that the importance of linear layers depends on the discretisation.

*Table 12.* Mean and Standard Deviation reported for PPG-DaLiA dataset with and without Linear layer in S$H^2$RFSSM Block

| Model | IM discretization | | | IMEX discretization | | |
|---|---|---|---|---|---|---|
| S$H^2$RFSSM | w/ lin w/o BN | w/ lin w/ BN | w/o lin w/ BN | w/ lin w/o BN | w/ lin w/ BN | w/o lin w/ BN |
| MSE | $11.8 \pm 0.9$ | $12.09 \pm 0.01$ | $11.27 \pm 0.67$ | $8.5 \pm 0.5$ | $10.08 \pm 0.46$ | $10.66 \pm 0.5$ |

### D.9. Kernel for Regression Tasks

Our oscillatory neurons act globally on the classification data. Spikes also tend to lose out on local information. To capture local features and provide continuous values, we provided our spike decoder with local context using a kernel regressor. We study its effectiveness on S$H^2$RFSSM w/o BN.

For very long-range sequences, to avoid oversmoothing while increasing the representation power of a linear decoder, we introduce convolution with a filter. This can be thought of as similar to an LI filter with learnable parameters. We investigate kernel size on the most energy efficient model S$H^2$RFSSM (IMEX) w/o lin w/ BN. From Table 13, we observe that for the 50k PPG-DaLiA dataset, a kernel size of 64 outperforms the 32 and 128 filters by an error of 0.005.

*Table 13.* Ablation on kernel size for regression task

| Model | S$H^2$**RFSSM-IMEX-w/o-Linear-w/-BN** | | |
|---|---|---|---|
| Kernel Size | 32 | 64 | 128 |
| MSE | $10.62 \pm 0.59$ | $10.08 \pm 0.46$ | $10.69 \pm 0.45$ |

We performed ablations on our best-performing regression model with IMEX discretisation. The kernel size controls the temporal receptive field of spike aggregation. For PPG sampled at 128 Hz, kernel sizes around 16–32 (approx 125–250 ms) align better with local waveform structures, leading to improved performance. In contrast, larger kernels (e.g., 64) tend to oversmooth the signal and show higher variability, as also observed in Figure 10. Such kernel sizes can be beneficial for learning with an IM discretisation due to its decaying nature (as observed in the best IM run; see Table 17).

As the sequence length increases (10k → 50k), sensitivity to kernel size reduces, but variance across runs increases, suggesting greater optimisation flexibility but also instability for larger kernels. Overall, the optimal kernel size is more strongly determined by signal time scales than by sequence length. This motivates the use of dynamic or multi-scale kernel strategies to balance local and global temporal dependencies.

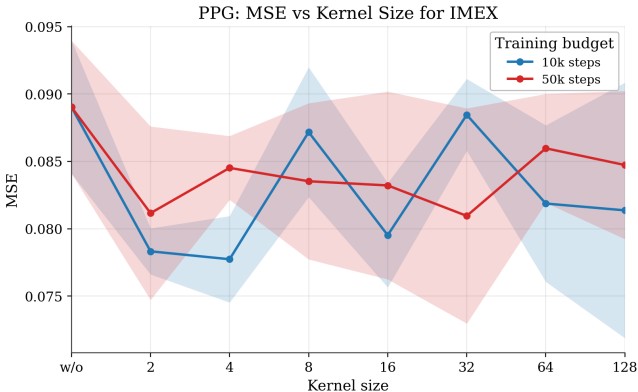

*Figure 10.* Ablation on the effect of kernel size on MSE for different training steps

## E. Extended Methods

### E.1. Implementation across architectures

Prior to state-space models, RNN-based architectures were the primary tools for long-sequence regression and classification. These models were later enhanced through discretisation (Rusch & Rus, 2025; Rusch & Mishra, 2021a;b), enabling stable discrete-time formulations. Building on this, we introduce an HRF layer to incorporate neuronal heterogeneity, improving expressivity and biological plausibility. For regression, rather than proposing a conventional encoder–decoder architecture, we position our method as a progression in sequence modelling architectures. We adopt the architectural principles of SpikF (Wu et al., 2025), replacing its S-FFT and S-iFFT components with our proposed $H^2$RFSSM. This substitution enables more effective signal processing while remaining neuromorphic-hardware friendly and computationally efficient, matching the performance of SpikF. However, due to limitations arising from SpikF's patch-based processing, this architecture proves insufficient for modelling long-range temporal interactions. To address this, we introduce a spiking non-linearity–based

*Table 14.* Statistics of long-horizon multivariate time-series forecasting benchmarks.

| Dataset | #Vars | Sampling | Train | Val | Test |
|---------|-------|----------|-------|-----|------|
| ECL | 321 | 1 Hour | 18,317 | 2,633 | 5,261 |
| Weather | 21 | 10 Minutes | 36,792 | 5,271 | 10,540 |
| ETTh | 7 | 1 Hour | 8,545 | 2,881 | 2,881 |
| ETTm | 7 | 15 Minutes | 34,465 | 11,521 | 11,521 |
| Traffic | 862 | 1 Hour | 12,185 | 1,757 | 3,509 |
| Exchange | 8 | 1 Day | 5,120 | 665 | 1,422 |

encoding mechanism (also studied via ablations). For S$H^2$RFSSM, we design an encoder that converts the input signal into spike trains for further processing. The encoder used for classification has slightly different components than the one used for regression, due to the task's complexity with long sequences. For classification, the encoder consists of a spiking nonlinearity, a linear projection, optional batch normalisation, and an integrate-and-fire (IF) neuron. The decoder is a simple linear layer, followed by a softmax operation for the final prediction. For PPG, a kernel-based regressor is used to decode spikes.

## F. Forecasting Benchmark

### F.1. Long-term Forecast

To test our model's capability for full-scale regression tasks, we compared it with state-of-the-art models. We replicated our results on the best hyperparameters as provided by Wu et al. (2025). We observe that, even without performing a hyperparameter search, our model can compete with SpikF without requiring carefully crafted FFT-based signal processing, which is implausible for neuromorphic hardware. HRF neurons can filter out relevant frequencies to forecast future time values.

### F.2. ETT Dataset

To test our model's capability for full-scale regression tasks, we compared it with state-of-the-art models. We replicated our results on the best hyperparameters as provided by (Wu et al., 2025). We observe that, even without performing a hyperparameter search, our model can compete with SpikF without requiring hand-crafted FFT-based signal processing. HRF neurons can filter out relevant frequencies to forecast future time values.

## G. Experimental Setup

We conducted our experiments on A6000 48 GB and A5000 24GB GPUs. Library versions were Jax-0.4.33, Numpy-1.26.4, Diffrax-0.5.1, Equinox-0.11.4, and Torch 2.8.0+cu128.

*Table 15.* MSE & MAE reported for all 8 long-term forecasting datasets for different prediction lengths=96,192,336,720

| Model | | SpikHRFSSM (Ours) | | SpikF (2025) | | iTransformer (2024) | | RLinear (2023) | | PatchTST (2023) | | Crossformer (2023) | | TimesNet (2023) | | DLinear (2023) | | SCINet (2022) | | Autoformer (2021) | |
|---|---|---|---|---|---|---|---|---|---|---|---|---|---|---|---|---|---|---|---|---|---|
| Metric | | MSE | MAE | MSE | MAE | MSE | MAE | MSE | MAE | MSE | MAE | MSE | MAE | MSE | MAE | MSE | MAE | MSE | MAE | MSE | MAE |
| ECL | 96 | 0.168 | 0.251 | 0.156 | 0.252 | 0.148 | 0.24 | 0.201 | 0.281 | 0.181 | 0.27 | 0.219 | 0.314 | 0.168 | 0.272 | 0.197 | 0.282 | 0.247 | 0.345 | 0.201 | 0.317 |
| | 192 | 0.178 | 0.26 | 0.169 | 0.262 | 0.162 | 0.253 | 0.201 | 0.283 | 0.188 | 0.274 | 0.231 | 0.322 | 0.184 | 0.289 | 0.196 | 0.285 | 0.257 | 0.355 | 0.222 | 0.334 |
| | 336 | 0.194 | 0.277 | 0.188 | 0.281 | 0.178 | 0.269 | 0.215 | 0.298 | 0.204 | 0.293 | 0.246 | 0.337 | 0.198 | 0.3 | 0.209 | 0.301 | 0.269 | 0.369 | 0.231 | 0.338 |
| | 720 | 0.239 | 0.313 | 0.219 | 0.306 | 0.225 | 0.317 | 0.257 | 0.331 | 0.246 | 0.324 | 0.28 | 0.363 | 0.22 | 0.32 | 0.245 | 0.333 | 0.299 | 0.39 | 0.254 | 0.361 |
| | Avg | 0.195 | 0.275 | 0.183 | 0.275 | 0.178 | 0.27 | 0.219 | 0.298 | 0.205 | 0.29 | 0.244 | 0.334 | 0.192 | 0.295 | 0.212 | 0.3 | 0.268 | 0.365 | 0.227 | 0.338 |
| Weather | 96 | 0.157 | 0.195 | 0.163 | 0.2 | 0.174 | 0.214 | 0.192 | 0.232 | 0.177 | 0.218 | 0.158 | 0.23 | 0.172 | 0.22 | 0.196 | 0.255 | 0.221 | 0.306 | 0.266 | 0.336 |
| | 192 | 0.206 | 0.242 | 0.209 | 0.241 | 0.221 | 0.254 | 0.24 | 0.271 | 0.225 | 0.259 | 0.206 | 0.277 | 0.219 | 0.261 | 0.237 | 0.296 | 0.261 | 0.34 | 0.307 | 0.367 |
| | 336 | 0.263 | 0.283 | 0.266 | 0.283 | 0.278 | 0.296 | 0.292 | 0.307 | 0.278 | 0.297 | 0.272 | 0.335 | 0.28 | 0.306 | 0.283 | 0.335 | 0.309 | 0.378 | 0.359 | 0.395 |
| | 720 | 0.342 | 0.336 | 0.344 | 0.334 | 0.358 | 0.347 | 0.364 | 0.353 | 0.354 | 0.348 | 0.398 | 0.418 | 0.365 | 0.359 | 0.345 | 0.381 | 0.377 | 0.427 | 0.419 | 0.428 |
| | Avg | 0.242 | 0.264 | 0.245 | 0.265 | 0.258 | 0.278 | 0.272 | 0.291 | 0.259 | 0.281 | 0.259 | 0.315 | 0.259 | 0.287 | 0.265 | 0.317 | 0.292 | 0.363 | 0.382 | 0.382 |
| ETTh1 | 96 | 0.381 | 0.392 | 0.379 | 0.391 | 0.386 | 0.405 | 0.386 | 0.395 | 0.414 | 0.419 | 0.423 | 0.448 | 0.384 | 0.402 | 0.386 | 0.4 | 0.654 | 0.599 | 0.449 | 0.459 |
| | 192 | 0.438 | 0.424 | 0.432 | 0.421 | 0.441 | 0.436 | 0.437 | 0.424 | 0.46 | 0.445 | 0.471 | 0.474 | 0.436 | 0.429 | 0.437 | 0.476 | 0.719 | 0.631 | 0.5 | 0.482 |
| | 336 | 0.475 | 0.444 | 0.473 | 0.441 | 0.487 | 0.458 | 0.479 | 0.446 | 0.501 | 0.466 | 0.57 | 0.546 | 0.491 | 0.469 | 0.481 | 0.541 | 0.778 | 0.659 | 0.521 | 0.496 |
| | 720 | 0.478 | 0.462 | 0.474 | 0.459 | 0.503 | 0.491 | 0.481 | 0.47 | 0.5 | 0.488 | 0.653 | 0.621 | 0.521 | 0.5 | 0.831 | 0.657 | 0.836 | 0.699 | 0.514 | 0.512 |
| | Avg | 0.443 | 0.431 | 0.44 | 0.428 | 0.454 | 0.447 | 0.446 | 0.434 | 0.469 | 0.454 | 0.529 | 0.522 | 0.458 | 0.45 | 0.559 | 0.515 | 0.747 | 0.647 | 0.496 | 0.487 |
| ETTh2 | 96 | 0.286 | 0.336 | 0.29 | 0.336 | 0.297 | 0.349 | 0.288 | 0.338 | 0.302 | 0.348 | 0.745 | 0.584 | 0.34 | 0.374 | 0.333 | 0.387 | 0.333 | 0.387 | 0.346 | 0.388 |
| | 192 | 0.361 | 0.384 | 0.367 | 0.385 | 0.38 | 0.4 | 0.374 | 0.39 | 0.388 | 0.4 | 0.877 | 0.656 | 0.402 | 0.414 | 0.477 | 0.476 | 0.86 | 0.689 | 0.456 | 0.452 |
| | 336 | 0.404 | 0.415 | 0.414 | 0.42 | 0.428 | 0.432 | 0.415 | 0.426 | 0.426 | 0.433 | 1.043 | 0.731 | 0.452 | 0.452 | 0.594 | 0.427 | 1 | 0.744 | 0.482 | 0.486 |
| | 720 | 0.42 | 0.438 | 0.416 | 0.436 | 0.427 | 0.445 | 0.42 | 0.44 | 0.431 | 0.446 | 1.104 | 0.763 | 0.462 | 0.468 | 1.249 | 0.838 | 1.249 | 0.838 | 0.515 | 0.511 |
| | Avg | 0.368 | 0.393 | 0.372 | 0.394 | 0.383 | 0.407 | 0.414 | 0.398 | 0.387 | 0.407 | 0.942 | 0.684 | 0.414 | 0.427 | 0.954 | 0.723 | 0.954 | 0.723 | 0.45 | 0.459 |
| ETTm1 | 96 | 0.318 | 0.348 | 0.317 | 0.345 | 0.334 | 0.368 | 0.355 | 0.376 | 0.329 | 0.367 | 0.404 | 0.426 | 0.338 | 0.375 | 0.345 | 0.372 | 0.345 | 0.372 | 0.505 | 0.475 |
| | 192 | 0.374 | 0.377 | 0.372 | 0.372 | 0.377 | 0.391 | 0.391 | 0.392 | 0.367 | 0.385 | 0.45 | 0.451 | 0.374 | 0.387 | 0.38 | 0.389 | 0.439 | 0.45 | 0.553 | 0.496 |
| | 336 | 0.399 | 0.396 | 0.401 | 0.394 | 0.426 | 0.42 | 0.424 | 0.415 | 0.399 | 0.41 | 0.532 | 0.515 | 0.41 | 0.411 | 0.413 | 0.413 | 0.49 | 0.485 | 0.621 | 0.537 |
| | 720 | 0.471 | 0.435 | 0.461 | 0.43 | 0.491 | 0.459 | 0.487 | 0.45 | 0.454 | 0.439 | 0.666 | 0.589 | 0.478 | 0.45 | 0.474 | 0.453 | 0.595 | 0.55 | 0.671 | 0.561 |
| | Avg | 0.39 | 0.389 | 0.388 | 0.385 | 0.407 | 0.41 | 0.414 | 0.407 | 0.387 | 0.4 | 0.513 | 0.496 | 0.4 | 0.406 | 0.403 | 0.407 | 0.485 | 0.481 | 0.588 | 0.517 |
| ETTm2 | 96 | 0.171 | 0.25 | 0.175 | 0.251 | 0.18 | 0.264 | 0.182 | 0.265 | 0.175 | 0.259 | 0.287 | 0.366 | 0.187 | 0.267 | 0.193 | 0.292 | 0.286 | 0.274 | 0.255 | 0.339 |
| | 192 | 0.238 | 0.294 | 0.242 | 0.296 | 0.25 | 0.309 | 0.246 | 0.304 | 0.241 | 0.302 | 0.414 | 0.492 | 0.249 | 0.309 | 0.284 | 0.362 | 0.399 | 0.445 | 0.281 | 0.34 |
| | 336 | 0.297 | 0.332 | 0.302 | 0.336 | 0.311 | 0.348 | 0.307 | 0.342 | 0.305 | 0.343 | 0.597 | 0.542 | 0.321 | 0.351 | 0.369 | 0.427 | 0.637 | 0.591 | 0.339 | 0.372 |
| | 720 | 0.391 | 0.39 | 0.405 | 0.397 | 0.412 | 0.407 | 0.407 | 0.398 | 0.402 | 0.414 | 1.73 | 1.042 | 0.408 | 0.403 | 0.554 | 0.522 | 0.96 | 0.735 | 0.433 | 0.432 |
| | Avg | 0.274 | 0.316 | 0.281 | 0.32 | 0.288 | 0.332 | 0.286 | 0.327 | 0.281 | 0.326 | 0.757 | 0.61 | 0.291 | 0.333 | 0.35 | 0.401 | 0.571 | 0.537 | 0.327 | 0.371 |
| Traffic | 96 | 0.47 | 0.281 | 0.477 | 0.286 | 0.395 | 0.268 | 0.649 | 0.389 | 0.462 | 0.295 | 0.522 | 0.29 | 0.593 | 0.321 | 0.65 | 0.396 | 0.788 | 0.499 | 0.613 | 0.388 |
| | 192 | 0.48 | 0.284 | 0.481 | 0.289 | 0.417 | 0.276 | 0.601 | 0.366 | 0.466 | 0.296 | 0.53 | 0.293 | 0.617 | 0.336 | 0.598 | 0.37 | 0.789 | 0.505 | 0.616 | 0.382 |
| | 336 | 0.497 | 0.291 | 0.499 | 0.295 | 0.433 | 0.283 | 0.609 | 0.369 | 0.482 | 0.304 | 0.558 | 0.305 | 0.629 | 0.336 | 0.605 | 0.373 | 0.797 | 0.508 | 0.622 | 0.337 |
| | 720 | 0.529 | 0.307 | 0.533 | 0.312 | 0.467 | 0.302 | 0.647 | 0.387 | 0.514 | 0.322 | 0.589 | 0.328 | 0.64 | 0.35 | 0.645 | 0.394 | 0.841 | 0.523 | 0.66 | 0.408 |
| | Avg | 0.494 | 0.291 | 0.497 | 0.296 | 0.428 | 0.282 | 0.626 | 0.378 | 0.481 | 0.304 | 0.55 | 0.304 | 0.62 | 0.336 | 0.625 | 0.383 | 0.804 | 0.509 | 0.628 | 0.379 |
| Exchange | 96 | 0.085 | 0.204 | 0.084 | 0.201 | 0.086 | 0.206 | 0.093 | 0.217 | 0.088 | 0.205 | 0.256 | 0.367 | 0.107 | 0.234 | 0.088 | 0.218 | 0.267 | 0.396 | 0.197 | 0.323 |
| | 192 | 0.181 | 0.303 | 0.18 | 0.3 | 0.177 | 0.299 | 0.184 | 0.307 | 0.176 | 0.299 | 0.47 | 0.509 | 0.226 | 0.344 | 0.176 | 0.315 | 0.351 | 0.459 | 0.3 | 0.369 |
| | 336 | 0.351 | 0.429 | 0.334 | 0.417 | 0.331 | 0.417 | 0.351 | 0.432 | 0.301 | 0.397 | 1.268 | 0.883 | 0.367 | 0.448 | 0.313 | 0.427 | 1.324 | 0.853 | 0.509 | 0.524 |
| | 720 | 0.935 | 0.732 | 0.841 | 0.69 | 0.847 | 0.691 | 0.886 | 0.714 | 0.901 | 0.714 | 1.767 | 1.068 | 0.964 | 0.746 | 0.839 | 0.695 | 1.058 | 0.797 | 1.447 | 0.941 |
| | Avg | 0.388 | 0.417 | 0.36 | 0.402 | 0.36 | 0.403 | 0.378 | 0.417 | 0.367 | 0.404 | 0.94 | 0.707 | 0.416 | 0.443 | 0.354 | 0.414 | 0.75 | 0.626 | 0.613 | 0.539 |

*Table 16.* ETT baseline for spike-based and transformer-based methods.

| Model | | SpikHRFSSM (Ours) | | SpikF (2025) | | SpikeRNN (2019) | | iSpikFormer (2024) | | FiTS (2024) | | FEDformer (2022) | |
|---|---|---|---|---|---|---|---|---|---|---|---|---|---|---|
| Metric | | MSE | MAE | MSE | MAE | MSE | MAE | MSE | MAE | MSE | MAE | MSE | MAE |
| ETTh1 | 96 | 0.381 | 0.392 | 0.379 | 0.391 | 0.392 | 0.395 | 0.41 | 0.408 | 0.386 | 0.396 | 0.376 | 0.419 |
| | 192 | 0.438 | 0.424 | 0.432 | 0.421 | 0.437 | 0.424 | 0.459 | 0.438 | 0.436 | 0.423 | 0.42 | 0.448 |
| | 336 | 0.475 | 0.444 | 0.473 | 0.441 | 0.482 | 0.447 | 0.514 | 0.461 | 0.478 | 0.444 | 0.459 | 0.465 |
| | 720 | 0.478 | 0.462 | 0.474 | 0.459 | 0.499 | 0.471 | 0.511 | 0.476 | 0.502 | 0.495 | 0.506 | 0.507 |
| | Avg | 0.443 | 0.431 | 0.44 | 0.428 | 0.452 | 0.434 | 0.473 | 0.446 | 0.451 | 0.44 | 0.44 | 0.46 |
| ETTh2 | 96 | 0.286 | 0.336 | 0.29 | 0.336 | 0.295 | 0.335 | 0.308 | 0.349 | 0.295 | 0.35 | 0.358 | 0.397 |
| | 192 | 0.361 | 0.384 | 0.367 | 0.385 | 0.375 | 0.387 | 0.382 | 0.397 | 0.381 | 0.396 | 0.429 | 0.439 |
| | 336 | 0.404 | 0.415 | 0.414 | 0.42 | 0.422 | 0.422 | 0.439 | 0.433 | 0.426 | 0.438 | 0.496 | 0.487 |
| | 720 | 0.42 | 0.438 | 0.416 | 0.436 | 0.428 | 0.438 | 0.438 | 0.439 | 0.431 | 0.446 | 0.463 | 0.474 |
| | Avg | 0.368 | 0.393 | 0.372 | 0.394 | 0.38 | 0.396 | 0.392 | 0.407 | 0.383 | 0.408 | 0.437 | 0.449 |
| ETTm1 | 96 | 0.318 | 0.348 | 0.317 | 0.345 | 0.326 | 0.351 | 0.337 | 0.361 | 0.355 | 0.375 | 0.379 | 0.419 |
| | 192 | 0.374 | 0.377 | 0.372 | 0.372 | 0.376 | 0.377 | 0.38 | 0.383 | 0.392 | 0.393 | 0.426 | 0.441 |
| | 336 | 0.399 | 0.396 | 0.401 | 0.394 | 0.396 | 0.394 | 0.415 | 0.406 | 0.424 | 0.414 | 0.445 | 0.459 |
| | 720 | 0.471 | 0.435 | 0.461 | 0.43 | 0.459 | 0.433 | 0.476 | 0.443 | 0.487 | 0.449 | 0.543 | 0.49 |
| | Avg | 0.39 | 0.389 | 0.388 | 0.385 | 0.389 | 0.389 | 0.402 | 0.398 | 0.415 | 0.408 | 0.448 | 0.452 |
| ETTm2 | 96 | 0.171 | 0.25 | 0.175 | 0.251 | 0.176 | 0.251 | 0.178 | 0.256 | 0.183 | 0.266 | 0.203 | 0.287 |
| | 192 | 0.238 | 0.294 | 0.242 | 0.296 | 0.243 | 0.296 | 0.243 | 0.298 | 0.247 | 0.305 | 0.269 | 0.328 |
| | 336 | 0.297 | 0.332 | 0.302 | 0.336 | 0.302 | 0.335 | 0.304 | 0.338 | 0.307 | 0.342 | 0.325 | 0.366 |
| | 720 | 0.391 | 0.39 | 0.405 | 0.397 | 0.409 | 0.396 | 0.406 | 0.398 | 0.407 | 0.399 | 0.421 | 0.415 |
| | Avg | 0.274 | 0.316 | 0.281 | 0.32 | 0.283 | 0.319 | 0.283 | 0.322 | 0.286 | 0.328 | 0.305 | 0.349 |

# H. Hyperparameters

*Table 17.* Best hyperparameters for each dataset and S$H^2$RFSSM model

| Dataset | Method | LR | Hidden | State | Blocks | Time | Drop | Kernel |
|---------|--------|------|--------|-------|--------|-------|------|--------|
| UCI-HAR | IM | 1e-3 | 128 | 256 | 2 | False | 0.0 | - |
|         | IMEX | 1e-3 | 128 | 256 | 2 | False | 0.0 | - |
| SHAR | IM | 1e-3 | 128 | 256 | 2 | False | 0.0 | - |
|      | IMEX | 1e-3 | 128 | 256 | 2 | False | 0.0 | - |
| Worms | IM | 1e-4 | 128 | 256 | 2 | False | 0.00 | - |
|       | IMEX | 1e-3 | 128 | 64 | 2 | False | 0.00 | - |
| SCP1 | IM | 1e-4 | 128 | 256 | 6 | False | 0.05 | - |
|      | IMEX | 1e-3 | 64 | 256 | 4 | True | 0.05 | - |
| SCP2 | IM | 1e-5 | 64 | 64 | 6 | True | 0.00 | - |
|      | IMEX | 1e-5 | 64 | 64 | 6 | True | 0.00 | - |
| Ethanol | IM | 1e-3 | 128 | 16 | 6 | True | 0.05 | - |
|         | IMEX | 1e-3 | 128 | 16 | 6 | True | 0.05 | - |
| Heartbeat | IM | 1e-5 | 16 | 256 | 2 | True | 0.05 | - |
|           | IMEX | 1e-3 | 64 | 16 | 4 | True | 0.05 | - |
| Motor | IM | 1e-5 | 128 | 16 | 4 | False | 0.00 | - |
|       | IMEX | 1e-5 | 128 | 16 | 2 | True | 0.05 | - |
| PPG | IM | 1e-3 | 64 | 64 | 4 | True | 0.00 | 64 |
|     | IMEX | 1e-3 | 128 | 256 | 6 | False | 0.00 | 16 |

