# OpenReview forum: "A Spiking Heterogeneous Harmonic Resonate-and-Fire State Space Model for Time Series"
_ICML.cc/2026/Conference — ICML 2026 regular_

### Official Review · Reviewer_rv2U · 2026-03-06

**Soundness:** 2
**Presentation:** 3
**Significance:** 3
**Originality:** 3
**Overall Recommendation:** 4
**Confidence:** 4

**Summary:**

This paper proposes SH2RFSSM, an energy-efficient second-order spiking State Space Model (SSM) built with harmonic resonate-and-fire (HRF) neurons and a learnable encoder and decoder. The model performs well on HAR datasets while consuming significantly less energy, and also achieves superior classification and regression performance on 18k-length EigenWorms and 50k-length PPG datasets, respectively.

**Compliance With Llm Reviewing Policy:**

Affirmed.

**Final Justification:**

After reading the authors' rebuttal and their responses to other reviewers, I believe my remaining concerns have been largely addressed. However, I still find the motivation somewhat thin, which is why I have not raised my score to 5.

**Key Questions For Authors:**

See weaknesses.

**Limitations:**

Yes

**Strengths And Weaknesses:**

Strengths

1. The authors conducted a substantial amount of experimental work, accompanied by numerous interesting and insightful analyses.

2. The proposed method demonstrates highly competitive results across a diverse and challenging range of datasets.

Weaknesses

1. The motivation lacks depth. The introduction mainly argues that HRF neurons are underexplored, but never clearly explains why existing spiking SSMs fail on very long sequences or how HRF neurons specifically address this gap.

2. The energy computation methodology is not presented in the main text, making it difficult to evaluate the validity of the reported efficiency claims.

3. Mathematical formulas are improperly formatted inline within the main text rather than being displayed as standalone, numbered block equations (e.g., line 109, right column, page 2).

---

> ### Author Rebuttal · Authors · 2026-03-31
>
> Thank you for taking the time to review our paper. Please find our replies below:
>
> **1. Motivation and Contribution**
> We would like to reiterate that the primary goal of this project is to achieve efficient target-variable modelling (i.e., accuracy and energy consumption) for long sequences. We observe that HRF neurons, with experimentally guided selection of heterogeneities, constitute a potent solution. Such architectures can be used on low-resource devices that require long sequence processing, such as battery-powered wearables.
>
> **2. Empirical findings not supported with scientific evidence**
> Thank you for your concern in this regard. Beyond empirical validation, our method explicitly grounds design choices in the dynamical system's properties. Fig. 2 shows how discretisation affects the system of continuous second-order dynamics, thereby directly affecting our parameterisation of the neuron. We also attempt to understand the spike patterns observed for the continuous system (Appendix B). Although heterogeneity is ubiquitous in biology, its role remains poorly understood. Previous works have studied heterogeneity in IF/LIF neurons. We here study the impact of heterogeneity on the more expressive R \& F neurons - with and without the SSM framework on appropriate datasets (please refer to reh3, yV5Y, 5dJF). Moreover, our experiments can act as a guide to the proper selection of heterogeneity and learnability for efficient performance (leading to reduced SOPs, etc.).
>
> **3. Inline equations to blocked equations**
> Thank you for your feedback. We would be happy to address this minor aspect in the final version.
>
> **4. Correcting Inaccuracies in the Review**
> We would like to point out a potential confusion. Our paper does not have any sections on the Continuous Wavelet Transform. Moreover, we have not experimented with SpokenArabicDataset. Please let us know if you want us to clarify anything.

---

> > ### Author Rebuttal · Reviewer_rv2U · 2026-04-04
> >
> > I apologize for mistakenly including two comments from a different paper in my original review. I have updated my weaknesses accordingly. After reading the authors' rebuttal and their responses to other reviewers, I believe my remaining concerns have been largely addressed. However, I still find the motivation somewhat thin, which is why I have not raised my score to 5.

---

> > > ### Author Response · Authors · 2026-04-07
> > >
> > > Thank you for reviewing our work.
> > >
> > > We elaborate here on the strengths of the proposed neuron dynamics and the connections between the parameterisation and long-range modelling.
> > >
> > > **HRF neurons are oscillating neurons which emulate phase properties similar to RF neurons, especially in the b=0 regime with a time period of $2\pi/\omega$**.
> > >
> > > Consider the second-order HRF neuronal dynamics:
> > > $$u'(t) = -\omega^2 v(t) - 2bu(t) + x(t),$$
> > > $$v'(t) = u(t),$$
> > >
> > > where $\omega > 0$, $b \in \mathbb{R}$, and $x(t)$ is an input. If we define the scaled coordinate $\tilde{v}(t) = \omega v(t)$, then the system can be equivalently written as
> > > $$u'(t) = -\omega \tilde{v}(t) - 2bu(t) + x(t),$$
> > > $$\tilde{v}'(t) = \omega u(t).$$
> > >
> > >
> > >
> > > **Theorem:** (Scaled Equivalence of Undamped HRF neurons and Undamped RF neurons)
> > > For $b = 0$, the dynamics reduce to
> > > $$w'(t) = i\omega w(t) + x(t), \qquad w(t) = u(t) + i \tilde{v}(t).$$
> > > which coincide with the polar form of the complex-valued first-order system of an undamped RF neuron.
> > >
> > >
> > > **Proof:**
> > > Set $b=0$ in the scaled system:
> > > $$u'(t) = -\omega \tilde{v}(t) + x(t), $$
> > > $$\tilde{v}'(t) = \omega u(t).$$
> > > Define the complex variable $w(t) = u(t) + i \tilde{v}(t)$. Then
> > > $$ w'(t) = u'(t) + i \tilde{v}'(t) $$
> > > $$\qquad= \big(-\omega \tilde{v}(t) + x(t)\big) + i(\omega u(t)) $$
> > > $$\qquad= i\omega (u(t) + i \tilde{v}(t)) + x(t) $$
> > > $$\qquad= i\omega w(t) + x(t).$$
> > > which matches the stated polar dynamics.
> > >
> > > $\square$
> > >
> > > **Remark:**
> > > The scaling $\tilde{v} = \omega v$ removes the anisotropy induced by $\omega$ in the original $(u,v)$ coordinates. Thus, the apparent discrepancy between the second-order real formulation and the first-order complex formulation arises purely from coordinate scaling.
> > >
> > > **Evolving dynamics of HRF neurons**
> > > Now, given the equivalence, we can emulate RF neurons with real-valued states and second-order dynamics. [From the phase diagram](https://anonymous.4open.science/r/ICML2026_Rebuttal-2678/phase_diagrams.png), we observe elliptical orbits due to the $\omega$ scaling.
> > >
> > >
> > >
> > > Such dynamics upon discretisation using IMEX or IM exhibit stability. IMEX discretisation also preserves oscillations, although they follow a different trajectory than in the continuous-time formulation. However, IM dynamics spiral down to the fixed point.
> > >
> > > In typical RNNs and SSMs, past information is encoded in hidden states that decay exponentially over time. In contrast, sustained oscillatory dynamics encode information in the state's phase and amplitude, preventing decay. Intuitively, this corresponds to information rotating in state space, allowing it to be preserved over long horizons.
> > >
> > > We will add these descriptions to our manuscript.
> > >
> > > > Why existing spiking SSMs fail on very long sequences
> > >
> > > Thank you for asking this question. State-of-the-art Spiking SSMs (pSpikeSSM[1], SpikingSSM[2]) augment Mamba-style SSMs with additional LIF neuron states for spiking memory. This enables them a stronger performance, however it inherits performance limitations from mamba and fails to outperform them. RF-based methods have been shown to outperform them on LRA benchmarks [3]. However, they also fall short of s4 baseline. We propose incorporating biologically plausible $\omega$-scaled undamped RF neuronal states, discretised using IM and IMEX, to capture stable ultra-long-range dependencies. In this undamped setting, the IMEX eigenvalues lie on the unit circle, thereby avoiding decay and enabling stable learning of long-range dependencies. For IM discretization, even under the largest observed settings ($\Delta t_{\max}=0.826$, $\Omega_{\max}=0.992$, $N=17984$ for the EigenWorms dataset; see [parameter distribution](https://anonymous.4open.science/r/ICML2026_Rebuttal-2678/data_dist.png)), the expected magnitude of the $N$-th power of the eigenvalues remains on the order of $10^{-4}$ (≈ $1.64 \times 10^{-4}$) [4], indicating that the signal does not vanish excessively even for long sequences directly benefiting the gradient flow through the spike functions adjusting the firing threshold.
> > >
> > > Lastly, we would surely add energy estimation (provided in Appendix G) and inline equations as suggested in the main manuscript.
> > >
> > > ---
> > > References
> > > 1. Bal, M., and A. Sengupta. "P-SPIKESSM: HARNESSING PROBABILISTIC SPIKING STATE SPACE MODELS FOR LONG-RANGE DEPENDENCY TASKS.", ICLR, 2025.
> > > 2. Shen, S., et al. “SpikingSSMs: Learning Long Sequences With Sparse and Parallel Spiking State Space Models”, AAAI 2025.
> > > 3. Zhang, D., et al. "Dendritic Resonate-and-Fire Neuron for Effective and Efficient Long Sequence Modelling." NeurIPS 2025.
> > > 4. Rusch, T. K., and D. Rus. "Oscillatory State-Space Models." ICLR 2025.

---

### Official Review · Reviewer_5dJF · 2026-03-12

**Soundness:** 2
**Presentation:** 2
**Significance:** 2
**Originality:** 2
**Overall Recommendation:** 3
**Confidence:** 2

**Summary:**

This paper introduces SH2RFSSM, a second-order spiking state-space architecture designed for modeling long sequential data. The model integrates Harmonic Resonate-and-Fire (HRF) neurons within a state-space framework, leveraging their real-valued second-order dynamics to represent oscillatory behavior. Compared with conventional LIF neurons, HRF neurons provide richer temporal representations and are therefore more capable of capturing long-range dependencies. They further introduce SpikHRFSSM, which replaces the FFT-based components of SpikF with HRF layers to handle multivariate long-term forecasting. Experimental validation is conducted across 17 datasets, including UEA time-series benchmarks, PPG-DaLiA regression, human activity recognition datasets, and widely used forecasting benchmarks.

**Compliance With Llm Reviewing Policy:**

Affirmed.

**Key Questions For Authors:**

Please refer to weaknesses.

**Limitations:**

No, I recommend the author to include this.

**Strengths And Weaknesses:**

Strengths
1. The integration of HRF neurons into an SSM framework is well-motivated, as oscillatory dynamics are a natural inductive bias for temporal sequence modelling, and the second-order formulation is more biologically grounded than LIF-based SSMs.
2. The 69x energy efficiency improvement over LinOSS on EigenWorms is compelling and well-supported with detailed energy computation methodology


Weaknesses
1. The heterogeneity study results in Table 1 show remarkably similar accuracy across almost all configurations (92.7–93.6%), with differences smaller than the reported standard deviations. It is hard to draw strong conclusions about what heterogeneity actually contributes, and the paper's narrative of heterogeneity being important is not compellingly supported by the numbers
2. The relationship between SH2RFSSM and SpikHRFSSM is underexplained. They appear to be quite different architectures sharing the HRF neuron, but the paper presents them somewhat interchangeably, creating confusion about what the core contribution actually is.
3. The energy consumption computation need more evidence to convince. Does this computation stem from real hardware(GPU, CPU or FPGA), or manually computed.

---

> ### Author Rebuttal · Authors · 2026-03-31
>
> Thank you for the insightful review. Below we address the concerns raised -
>
> **1. Motivation behind the Heterogeneity study**
> Thanks for asking this question! Heterogeneity is ubiquitous in biology. For our biology-inspired [SH2RFSSM], the goal here is to not only understand the benefits but also observe the model robustness under heterogeneity. Previous studies have examined heterogeneity in IF/LIF neurons only. We here study the impact of heterogeneity in the more expressive HRF neurons for inference.  Higuchi et al. observed 92.7 ± 0.7% and 4139.5 SOPs using a BHRF neuron. We performed a detailed grid search on all heterogeneous neuronal parameters $(u,b,\omega,\theta,\Delta t,v)$ (Refer to Appendix Figure 7). We observe that heterogeneity and discretisation can improve model performance while significantly reducing SOPs. For example, using BHRF with IM discretisation results in a 0.7\% improvement in accuracy, with SOPs reduced to 2937.7, corresponding to 1.4$\times$ lower SOPs (Table 1). Hence, we conclude that neuronal heterogeneity and discretisation can directly impact model performance, both in terms of accuracy and SOPs, which in turn translate into energy efficiency. We observed similar trends in our study in an SSM framework (see the table below). Hence, we conclude that heterogeneity and discretisation affect overall performance not only in terms of accuracy but also in SOPs, which directly translate into energy consumption.
>
> **2. Energy consumption**
> We here follow the well-accepted energy estimation approach adopted in literature and as detailed in [refer to appendix sec G]. We believe this approach should also guide the efficient design of future production-ready neuromorphic hardware.
>
> Below, we attempt to highlight the central theme of our paper, energy efficiency and performance benefits by means of heterogeneity in neuronal parameters and discretisation (similar to the study conducted on SHD in Table 1).
>
> | Model |  u, b, $\Omega$, $\theta$, $\Delta$t, v | Accuracy | SOPs$\times 10^6$ | Energy | Steps$\times 10^3$
> | :--- | :--- | :---: | :---: | :---: | :---: |
> | **IMEX** | ✗ ✗ ✓ ✓ ✓ ✗ | 90.0 $\pm$ 5.72 | 365.53 $\pm$ 7.17 | 69.22× | 22.4 $\pm$ 4.4 |
> | | ✗ ✓ ✓ ✓ ✓ ✗ | 88.33 $\pm$ 4.78 | 281.15 $\pm$ 28.47 | 85.83× | 24.6 $\pm$ 7.1 |
> | | ✓ ✗ ✓ ✓ ✓ ✓ | 91.11 $\pm$ 4.78 | 362.06 $\pm$ 12.04 | 69.78× | 23.0 $\pm$ 4.5 |
> | | ✓ ✓ ✓ ✓ ✓ ✓ | 88.33 $\pm$ 6.18 | 299.97 $\pm$ 27.77 | 81.47× | 23.4 $\pm$ 6.8 |
> | **IM** | ✗ ✗ ✓ ✓ ✓ ✗ | 91.11 $\pm$ 3.68 | 403.51 $\pm$ 14.92 | 63.68× | 17.6 $\pm$ 3.0 |
> | | ✓ ✗ ✓ ✓ ✓ ✓ | 91.11 $\pm$ 4.78 | 395.17 $\pm$ 12.63 | 64.82× | 17.4 $\pm$ 6.1 |
>
> We observe an accuracy v/s SOPs tradeoff. Additional learnable parameters $(u,v,b)$ originally set to zero in the HRF layer of S$H^2$RFSSM on EigenWorms (18k sequence). We compared Energy consumption with respect to LinOSS on the same hyperparameter combination. The same hyperparameter was used as in the best-performing IMEX model, resulting in 10\% better performance than LinOSS-IMEX (see Table 7) across both discretisations for a fair comparison. b learnable initalized heterogeneously. We observe that learnable u and v can help capture phase information and improve overall model performance. However, he models with learnable b may lower overall performance, as observed, and rarely improve performance (Refer to reh3 for more insights). ($\Omega, \theta,\Delta t$) are further analysed in the main paper (Figure 5). IM discretisation requires fewer steps to converge to optimal accuracy, as we also observed in Table 2.
>
> **3. Clarification on HRF layer generalisation across architectures.**
> Sure. Happy to clarify on this. Our core contribution is: `A fully spiking state space model (SH2RFSSM) for classification and regression tasks, along with a detailed study on leveraging discretisation and heterogeneity in proposed HRF layers. In addition, we also explored HRF layers as a replacement to FFTs (SpikHRFSSM) in a state-of-the-art SNN architecture, SpikF (which are computationally expensive) for long sequence forecasting tasks.`
>
> We will update the paper with this statement. Please let us know if you have any further concerns.

---

> > ### Author Rebuttal · Reviewer_5dJF · 2026-04-03
> >
> > The rebuttal provides some clarification, especially on the distinction between SH2RFSSM and SpikHRFSSM, but I do not think it fully resolves my main concerns. In particular, the evidence for the importance of heterogeneity remains limited, and the energy analysis still relies on estimation rather than clearly validated hardware measurements.

---

> > > ### Author Response · Authors · 2026-04-07
> > >
> > > Thank you for your feedback. We took the best results from Table 2 as per the hyperparameters and additionally decoupled the effect of heterogeneity in SH2RFSSM below:
> > >
> > > |u, b, $\Omega$, $\theta$, $\Delta$t, v|Worms |SCP1 | SCP2 | Ethanol | Heartbeat | Motor| Average |
> > > | :--- | :--- | :---: | :---: | :---: |  :---: | :---:| :---: |
> > > |✗ ✗ ✗ ✗ ✗ ✗ | 75.6 $\pm$ 17.3 | 79.1 $\pm$ 3.4 | 52.9 $\pm$ 6.1 | 31.9 $\pm$ 2.6 | 75.8 $\pm$ 4.9 | 55.1 $\pm$ 5.2| 61.7 |
> > > |✗ ✗ ✓ ✓ ✓ ✗ | 92.8 $\pm$ 3.3 | 84.2 $\pm$ 3.2 | 59.3 $\pm$ 7.7 | 34.7 $\pm$ 7.1 | 74.5 $\pm$ 3.4 | 59.6 $\pm$ 5.4| 67.5 |
> > > |✓ ✗ ✓ ✓ ✓ ✓  | 92.6 $\pm$ 3.5 | 79.1 $\pm$ 3.5 | 58.9 $\pm$ 5.2 | 32.6 $\pm$ 4.8 | 76.1 $\pm$ 2.1 | 56.5 $\pm$ 5.1|65.9|
> > >
> > > We compare the inference accuracy between a homogeneous and a heterogeneous SH2RFSSM model. Notably, model heterogeneity can be introduced by initialisation alone, by learning alone, or by both. Hence, for a fair homogeneous baseline, u,v, and b are set to zero, $\Delta t$ is set to 0.01, $\theta$ to 1 and $\Omega$ is set with a unity trace. None of them undergoes learning.
> > >
> > > **We observe that a variant of the heterogeneous model always outperforms the homogenous model**. Moreover, heterogeneity across all is not necessarily better for performance (65.9\% for all heterogeneity vs. 67.5\% for a subset). Specifically, we observe that all heterogeneous parameters improved heartbeat performance the most on the smallest dataset in the benchmark. We did not include the b parameter due to the damping nature of IM discretisation and its inefficacy in previously reported results.
> > >
> > > We summarise our studies on heterogeneity as follows for different parameter learnable regimes:
> > > * For HRF in SSM setup (SH2RFSSM),
> > >     * Without heterogeneity, though we observe good performance for the smallest length heartbeat dataset, the homogeneous version on average only achieves 61.7\%. On the subsequently longer sequence datasets: EigenWorms, SCP1 and SCP2,  model heterogeneity delivers a significant performance boost(67.5\% on average).
> > >     * $(\Omega,\theta,\Delta t)$ remains the most desired heterogeneity across datasets.
> > >     * $(b, \Omega,\theta,\Delta t)$ heterogeneity offers dataset-specific boost in performance for SCP1. This selective performance improvement was also observed in [1].
> > >     * $(u,\Omega,\theta,\Delta t, v)$ heterogeneity is observed to reduce SOPs (first rebuttal, 5dJF); however, it is not necessarily always the optimal combination across datasets as observed in both paradigms. Hence, application-specific selection of heterogeneity is important.
> > >
> > > * For HRF in RNN setup (BHRF),
> > >     * We observe no benefit of using a learnable $\Delta t$.
> > >     * $(u,b,\omega,v)$ heterogeneity is the most optimal combination and under IM, significantly reduces SOPs to 2937.7 from reported results of 4139.5 and 0.7% improvement in accuracy over baseline, IMEX resulted in reduced SOPs $2991.1$ and higher error bars which is in coherence to classification performance in SSM setup where we observed IM suits classification tasks.
> > >     * $(u,b,\omega,\theta,v)$ heterogeneity results in reduced SOPs in general, but not the most optimal combination. Hence, learning all parameters is not guaranteed to improve performance.
> > >
> > > We will update the manuscript with these additional results.
> > >
> > > > Energy analysis still relies on estimation rather than clearly validated hardware measurements.
> > >
> > > Thanks for your concern! We would like to point out that our energy estimation closely follows hardware behaviour, as is well accepted and used in recent SNN deep learning works [2-5]. Hence, we believe, given the extensive studies in this work, a direct hardware implementation is beyond the scope of this paper. This work should guide the optimal design of neuromorphic hardware. Moreover, 1st-order SSM and RF neurons have already been shown to be efficient on Intel Loihi 2[6]. In fact, unlike prior spiking SSM baselines, including pspikeSSM[4], we ensure high performance without the ANN non-linearities, thereby ensuring a fully SNN architecture. In addition, the use of biologically plausible HRF neurons in SH2RFSSM can help bridge the gap between deep learning and neuroscience, enabling better explanations of brain computations.
> > >
> > > ---
> > > References
> > > 1. Boyer, J., T. K. Rusch, and D. Rus. "Learning to Dissipate Energy in Oscillatory State-Space Models." arXiv preprint arXiv:2505.12171 (2025).
> > > 2.  Zhou, Z., et al. Spikformer: When spiking neural network meets transformer. ICLR, 2023.
> > > 3. Lee, D., et al. "Spiking transformer with spatial-temporal attention.", CVPR, 2025.
> > > 4. Bal, M., and A. Sengupta. "P-SPIKESSM: HARNESSING PROBABILISTIC SPIKING STATE SPACE MODELS FOR LONG-RANGE DEPENDENCY TASKS.", ICLR, 2025.
> > > 5. Shen, S., et al. “SpikingSSMs: Learning Long Sequences With Sparse and Parallel Spiking State Space Models”, AAAI 2025.
> > > 6. Orchard, G., et al. "Efficient neuromorphic signal processing with Loihi 2." 2021 IEEE workshop on signal processing systems (SiPS). IEEE, 2021.

---

### Official Review · Reviewer_yV5Y · 2026-03-15

**Soundness:** 3
**Presentation:** 3
**Significance:** 2
**Originality:** 2
**Overall Recommendation:** 4
**Confidence:** 3

**Summary:**

This work proposes a harmonic resonate and fire spiking neural network model based on second-order SSMs such as LinOSS. The model is competitive across a range of classification and regression tasks compared to other recent spiking and non-spiking sequence models. The authors further investigate numerous aspects of the model design, including discretization, cellular heterogeneity, and more.

**Compliance With Llm Reviewing Policy:**

Affirmed.

**Final Justification:**

Technically solid paper that advances spiked-based AI, with a contribution that others are likely to build on, though in a relatively narrow domain and with a marginal gain in performance (though noting the efficiency argument).

**Key Questions For Authors:**

1. Why is there an inconsistent choice of baseline methods compared across the different types of tasks, as well as their visualization (e.g., Table 2 & 3 vs. Figure 4.)? Can all models be run on all tasks, or are the numbers reported for the other models taken from published results?
2. Could you clarify Table 4 and the heterogeneity experiment? I’m having trouble parsing which is the “standard” model and whether the table shows ablation or addition of the various heterogeneities? It appears that network without batchnorm and homogeneous neurons (last row) performs best on this task, so why are the heterogeneity necessary?

**Limitations:**

There lacks a discussion of assumptions and limitations of this work.

**Strengths And Weaknesses:**

Soundness:
The proposed model appears adequately designed, with comprehensive evaluation experiments as well as analyses and ablation studies. It’s based on a combination of SNN and SSM models while also drawing from neuroscience literature such as heterogeneous spiking networks. I sometimes found the choice of analyses beyond the benchmarking results to be a little confusing and not well motivated.

Presentation:
I found the paper to be generally well written, with an especially clear motivation and survey of the background literature. The presentation of experiments and results is adequate, though I’m confused by the choice of figures (bar plots) vs. tables to present similar information across different task types. The visualizations are generally legible and well designed, though fonts are sometimes very small, and coloring of the top 3 models makes it difficult to see.

Significance:
I think this is an interesting work bringing together SSMs and second order SNNs that can inspire future works in similar domains. However, benchmark performance of the proposed SH2RFSSM is not so great, calling into question the necessity of the formulation proposed here, as well as SNNs in general for these tasks, though compared against SNN models it is competitive.

Originality:
I don’t know the SNN literature very well, but the current work appears to be a straightforward application of concepts from SSMs to SNNs, in particular when framed as turning LinOSS into a spiking model. It does draw on prior work on neuronal heterogeneity, which I thinks is interesting.

---

> ### Author Rebuttal · Authors · 2026-03-31
>
> Thank you for your detailed review. The primary goal of SNN network design is to achieve energy efficiency comparable to that of ANN-based methods while closely matching their performance. This is achieved by realising a spike-based architecture implementation instead of matrix multiplication in ANNs [1,2,3]. Presently, there is a significant requirement for such SNN-based Deep Learning architectures. We employ a fully spike-based spiking neural network without utilising GeLU/GLU non-linearities (Refer to the Section below).
>
> **1. Baseline clarification**
> Thank you for asking for clarification.
>
> For initial experiments, we utilised the relatively simple Spiking Heidelberg Dataset (SHD) to understand heterogeneity and discretisation to improve accuracy and energy efficiency (measured by the Number of Spiking Operations, i.e., SOPs incurred) in simple SNNs.
>
>
> Next, we proposed an HRF layer within an SSM framework to model ultra-long sequence classification and regression tasks and compared it with baselines from the State-of-the-art [2,3], achieving minimal performance degradation while achieving massive energy efficiency. There have been many recent papers that have positioned their work for long sequence classification and regression tasks (up to 18k sequences)[4,5]. These works emphasised a boost in performance on both long-sequence classification and regression tasks (up to 50 sequences)[6,7,8]. In the Tables 2 & 3 we attempt to compile all these results into one table to fairly compare our model to the state of the art models. Figure 4 results are taken from the state of the art on HAR datasets, namely UCI-HAR and SHAR[9]. We conducted experiments on state-of-the-art spiking SSM baselines [2,3] for fair comparison.
>
> **2. Motivation behind SNN for regression**
> We employ SNNs for forecasting in settings where the target function arises from an underlying dynamical system, particularly under sparsity and energy constraints. Our oscillatory SNN captures frequency-specific structure through its intrinsic dynamics, enabling effective modelling of periodic signals. Additionally, its event-driven computation offers improved energy efficiency, albeit with a modest trade-off in predictive accuracy compared to dense architectures. In this work, we present HRF layers for regression tasks as a replacement for FFT-IFFT layers in SpikF[1], which are computationally expensive.
>
> **3. Ablation and Heterogeneity clarification**
>
> We sincerely thank you for letting us know of the confusion. We have decoupled them into 2 tables on the heterogeneity study and model ablation tables.
> |Parameters |Heterogeneity |Set Value |Accuracy|
> | :--- | :--- | :---: | :---: |
> |$\theta$ Encoder |U(0, 1] |1.0 |85.0 $\pm$ 7.2|
> |$\Omega$| U(0, 1] |1.0 |82.2 $\pm$ 3.3|
> |B| U($-H^∗, H^∗$) |0.0 |85.0 $\pm$ 4.5|
> |C| U($−P^∗, P^∗$) |0.0 |85.0 $\pm$ 4.5|
> |D| N (0, 1) |0.0 |80.5 $\pm$ 4.6|
> |$\Delta$t| U(0, 1] |1.0 |82.8 $\pm$ 2.7|
> |$\theta_C$| U(0, 1] |1.0 |83.3 $\pm$ 3.9|
> |$\theta_D$| U(0, 1] |1.0 |85.0 $\pm$ 4.8|
> |$\Omega$,B,C,D |As Above |As Above |85.5 $\pm$ 4.1|
> |$\theta_C,\theta_D$|As Above |As Above |83.9 $\pm$ 5.1|
> |All |As Above |As Above |85.5 $\pm$ 4.1|
>
> Studying the role of heterogeneously initialised neuron $(\Omega,\theta,\Delta t)$ and SSM parameters (B, C, D) in an HRF-SSM Block on the EigenWorms dataset for five seeds on the IMEX (w/ BN) model (Here, $x^{*} =x^{-1/2}$).
>
> Empirically, we observe that heterogeneous initialisation improves accuracy and narrows error bars. To further emphasise our claim that heterogeneity benefits model performance in an SSM setup, we conducted additional experiments to understand the model's benefits in terms of accuracy and energy efficiency (Refer to 5dJF). As sequence length grows, we also observe that w/o BN variants perform better (please refer to reh3).
>
>
> |Model Components| Accuracy|
> | :--- | :---: |
> |w/ BN| 85.6 ± 2.7|
> |w/ BN (Only SSM in Block)| 87.2 ± 3.3|
> |w/o BN| 90.0 ± 5.7|
> |w/o BN w/ GeLU| 92.2 ± 5.1|
>
> Ablation of S$H^2$RFSSM-IMEX components on the EigenWorms dataset.
>
> Although the best results use GeLU/GLU non-linearities in the SSM block, we adopt a fully spike-based variant (S$H^2$RFSSM without batch normalisation) as the baseline. For classification, we observed IM discretisation performs better (shown to converge in fewer time steps in 5dJF heterogeneity table), hence, we achieve 92.8% (please refer to Table 2, main paper).
>
> Please let us know if you have any other suggestions.

---

> > ### Author Rebuttal · Reviewer_yV5Y · 2026-04-03
> >
> > Thank you for the clarifications, which improved the consistency and resolved some confusion on my part. The additional experiments are insightful, though I still have reservations about the significance / breadth of this work. Therefore I will raise my score to 4, recommending weak acceptance.

---

> > > ### Author Response · Authors · 2026-04-07
> > >
> > > Thanks for promising to raise the present score!
> > >
> > > A prime goal of neuromorphic computing is to attain energy efficiency by understanding and mimicking the brain. In-depth studies on Intel Loihi/Loihi 2 have clearly shown that RNNs (and hence SSMs, which are just their more expressive versions) are specifically suited for such efficient hardware implementations[1]. We believe the significance lies in handling ultra-long sequences with energy efficiency. We justify the model's 69% energy efficiency with comparable performance on the longest time-series datasets. Such architectures can be used on low-resource devices that require long sequence processing, such as battery-powered wearables. We demonstrate the efficacy of our model across up to 50k sequence lengths, achieving enormous energy savings.
> > >
> > > Due to the high demand for efficient AI, there has recently been significant interest and progress in designing SNN deep learning architectures [2-5].  However, most of these designs use an LIF/IF neuron. Instead, we here study the more expressive Resonate and Fire neurons, which have been widely accepted as better representing the Hodgkin-Huxley (HH) neuron behaviour despite being computationally simpler than the HH neuron. Moreover, by using RF neurons instead of LIF/IF neurons, this framework significantly reduces the gap between deep learning and neuroscience, potentially helping to understand the brain. Empirically, we show that HRF neurons in an SSM framework are well suited for long-sequence classification and regression [SH2RFSSM] as well as for signal processing [SpikHRFSSM]. To the best of our knowledge, the efficient performance on such diverse tasks, as demonstrated by SH2RFSSM, has not been reported earlier.
> > >
> > >
> > > RF neurons provide a unifying mechanism for frequency-selective, time-domain processing, enabling end-to-end neuromorphic solutions that eliminate spectral preprocessing while achieving high accuracy and ultra-low power across speech, radar, and biomedical applications [6-9]. We highlight these application areas in our work by showcasing consistently low MSE and MAE scores and equivalent performance to spectral methods (SpikF), which are seemingly infeasible on neuromorphic hardware. For instance, the Electricity dataset contains strong periodic patterns and high multivariate correlations demand time–frequency features; the Weather dataset contains seasonal/periodic patterns with low-dimensional structure suiting time–frequency representations; the Traffic dataset contains multi-scale patterns and sensor synchronisation benefitting from time–frequency modeling; and the Exchange dataset exhibits non-stationary, evolving frequencies calling for time–frequency analysis.
> > >
> > > Overall: These datasets naturally demand time–frequency representations, and our approach achieves this without explicit spectral preprocessing, by learning frequency-selective behaviour directly in the time domain. In embedded AI applications, this will enable replacing energy-intensive FFT accelerators with our HRF-based design, thereby significantly improving efficiency, as empirically demonstrated.
> > >
> > > We believe this is a significant contribution to the field.
> > >
> > > You may also refer to the insights we have provided on heterogeneity(5dJF) and the effectiveness of HRF neurons for long sequences (rv2U) in our rebuttal.
> > >
> > > References
> > > 1. Meyer, S. M., et al. "A diagonal state space model on Loihi 2 for efficient streaming sequence processing." NeurIPS 2024 Workshop Machine Learning with new Compute Paradigms.
> > > 2. Zhou, Z., et. al Spikformer: When spiking neural network meets transformer. ICLR, 2023.
> > > 3. Lee, D., et al. "Spiking transformer with spatial-temporal attention.", CVPR, 2025.
> > > 4. Bal, M., and A. Sengupta. "P-SPIKESSM: HARNESSING PROBABILISTIC SPIKING STATE SPACE MODELS FOR LONG-RANGE DEPENDENCY TASKS.", ICLR, 2025.
> > > 5. Shen, S., et. al “SpikingSSMs: Learning Long Sequences With Sparse and Parallel Spiking State Space Models”, AAAI 2025.
> > > 6. Orchard, G., et al. "Efficient neuromorphic signal processing with loihi 2." 2021 IEEE workshop on signal processing systems (SiPS). IEEE, 2021.
> > > 7. Hille, J., et al. "Resonate-and-fire neurons for radar interference detection." Proceedings of the International Conference on Neuromorphic Systems 2022. 2022.
> > > 8. O’Leary, G., et al. "BrainForest: Neuromorphic multiplier-less bit-serial weight-memory-optimized 1024-tree brain-state classification processor." IEEE Transactions on Biomedical Circuits and Systems 19.1 (2024): 55-67.
> > > 9. Manna, D. L., T. Bihl, and G. D. Caterina. "Resonate-and-fire neurons meet EMG: enhancing gesture classification with spiking neural networks." IEEE ICASSP, 2026.

---

### Official Review · Reviewer_reh3 · 2026-03-16

**Soundness:** 3
**Presentation:** 3
**Significance:** 3
**Originality:** 2
**Overall Recommendation:** 4
**Confidence:** 2

**Summary:**

This paper proposes  SH^22RFSSM, a second-order spiking state space model (SSM) using harmonic resonate-and-fire (HRF) neurons for ultra-long time series classification, regression, and forecasting. By integrating neuronal heterogeneity and stable discretisation schemes (IM/IMEX), the model eliminates matrix multiplications and achieves superior energy efficiency (~69× vs. LinOSS) while outperforming transformers and first-order SSMs on average. Extensions include a kernel-based regressor for 50k-step sequences and SpikHRFSSM for multivariate forecasting. This research appears to investigate the concept of merging spiking neural network dynamics with SSM scalability to address edge AI constraints. This article claims to investigate a major issue: the trade-off between expressivity, scalability, and energy efficiency in long-sequence modelling for resource-constrained devices.

**Compliance With Llm Reviewing Policy:**

Affirmed.

**Final Justification:**

The authors delivered detailed responses that have completely resolved all the concerns raised in my initial review. On this basis, I have determined my final score.

**Key Questions For Authors:**

1.For the heterogeneous HRF neuron parameters (ω, b, thresholds, Δt), what specific initialisation distributions/ranges were used across datasets, and how were these choices justified (e.g., biological plausibility, empirical tuning)? Were there dataset-specific patterns in optimal heterogeneity settings?
2.The kernel-based spiking regressor achieves optimal performance with a kernel size of 64 on PPG-DaLiA—what is the intuition behind this kernel size choice, and how does kernel size scale with sequence length (e.g., 10k vs. 50k steps)? Is there a dynamic kernel size strategy that could further improve regression performance?
3. SpikHRFSSM replaces FFT/IFFT layers in SpikF with HRF layers for forecasting—how does the frequency resolution of HRF neurons compare to FFT-based methods, and why does HRF-based filtering outperform FFT on some datasets (e.g., Weather, ETTh2) but not others (e.g., Exchange)?

**Limitations:**

Yes.

**Strengths And Weaknesses:**

Strengths:
1.Novel fully spike-based second-order SSM design, filling gaps in spiking models for ultra-long sequences.
2.Comprehensive analysis of discretisation and heterogeneity effects on HRF neurons, providing actionable insights for SNN design.
3.Extensive validation across 17 datasets (classification, regression, HAR, forecasting) with consistent performance gains over SOTA.
Weaknesses:
1.Limited Ablation on Model Components: The ablation study on linear layers in SSM blocks is inconclusive, and there is little analysis of how individual components (e.g., kernel regressor size, spike encoder design) contribute to performance on different task types (e.g., short vs. ultra-long sequences).
2.Heterogeneity Initialisation Details: The paper describes introducing heterogeneity in HRF neuron parameters (ω, b, thresholds) but provides limited details on the initialisation strategies for heterogeneous parameters and how they are tuned across datasets, making reproducibility challenging.
3.Figure 6 suffers from poor readability due to overly small dimensions; key details of pre/post-HRF spike patterns and learned frequencies are obscured.
4. Figure 1 has an unappealing and poorly formatted legend; visual elements and notation explanations lack clarity and aesthetic consistency.

---

> ### Author Rebuttal · Authors · 2026-03-31
>
> Thank you for your thorough review.
>
> We would like to emphasise that prior spiking SSM works [1] indicate a steep drop in performance when GeLU non-linearities are removed from their designs. Whereas the performance of our implementation is not particularly sensitive, it is completely spike-based [yV5Y].
>
> Please find our clarifications below:
>
> **1. Clarification regarding ($\omega, b, \theta, \Delta t$) initalization distributions**
> Since we operate on ultra-long sequences, maintaining stable training over long timesteps is critical. For continuous neurons, stable oscillations occur when $0 \leq b < \omega$, though the exact stability region depends on the discretisation (Fig. 2). For IM and IMEX schemes, stability is preserved within this range.
> For longer sequences, we omit $b$ and initialize $\Omega \sim U(0,1)$ and $\Delta t \sim U(0,1)$ (Rusch & Rus et al., 2025). We also set $\theta \sim U(0,1)$, allowing each HRF neuron to learn its own firing threshold.
> We observe the following data-specific insights:
>
> https://anonymous.4open.science/r/ICML2026_Rebuttal-2678/data_dist.png
>
> The input mixing vector (D) shows the highest variability, highlighting its key role in adapting spike injection to temporal structure, while B and C remain concentrated, indicating stable spike transformations.
>
> The threshold ($\theta$) stays within [0,1] but shows higher variance for EthanolConcentration, suggesting benefits of heterogeneous firing under noisy dynamics. The time-step ($\Delta t$) remains close to 1, supporting stable gradients under IM/IMEX discretisations.
>
> Stability is governed by $\Omega$ and b ($b < \Omega$), with b typically set to zero for stable long-range learning. Notably, SelfRegulationSCP1 benefits from a learnable b, consistent with [2]. Finally, Batch Normalisation improves performance for SCP1 and MotorImagery by capturing data-specific patterns.
>
> **2. Ablations on spike encoder design**
> We previously conducted normalisation ablations (Table 7) and highlight the following observations:
>
> |Spike Encoder |Worms (17984) |SCP2 (1152) |Heartbeat (405)|
> | :--- | :--- | :---: | :---: |
> |w/ LN| 79.4 $\pm$ 17.3 | 51.9 $\pm$ 6.9|73.2 $\pm$ 2.2|
> |w/ BN |85.6 $\pm$ 8.1|55.4 $\pm$ 6.9 |74.1 $\pm$ 5.9|
> |w/o Norm |92.8 $\pm$ 3.3|59.3 $\pm$ 7.7|74.5 $\pm$ 3.|
>
> We observe a clear dependence on sequence length. For shorter sequences, BN provides modest gains, likely by stabilising activation scale early in training. In contrast, for longer sequences, removing normalisation consistently improves both accuracy and variance. We hypothesise that, in this regime, the structured state-space dynamics, together with heterogeneous per-channel thresholds, are sufficient to regulate signal scale over time, rendering explicit normalisation unnecessary. Moreover, BN/LN introduce input-dependent rescaling of pre-threshold activations, which can interfere with the learned thresholds and disrupt temporally consistent spike generation.
>
> **3. Ablations on Kernel-based spiking regressor**
> https://anonymous.4open.science/r/ICML2026_Rebuttal-2678/ppg_sensitivity.png
>
> We thank the reviewers for the suggestion. We perform ablations (3 runs) on our best IMEX-based regression model to investigate the effect of kernel size. The kernel controls the temporal receptive field of spike aggregation. For PPG at 128 Hz, kernel sizes of 16–32 (~125–250 ms) align well with local waveform structure, yielding better performance. Larger kernels (e.g., 64) oversmooth the signal and show higher variance, as seen in the figure. However, such kernels can be beneficial under IM discretisation due to their decaying dynamics (best IM run; Table 13).
>
> As sequence length increases, sensitivity to kernel size decreases, while variance across runs increases, indicating greater optimisation flexibility but reduced stability for larger kernels. Overall, the optimal kernel size is more governed by signal time scales than by sequence length, motivating adaptive or multi-scale kernels to balance local and global dependencies.
>
> **4. Connection between FFT and HRF Layers**
> We thank the reviewers for their insightful comments. Our goal is not to outperform SpikF through extensive tuning, but to investigate whether explicit spectral transforms are necessary for frequency modelling. By replacing S-FFT/S-iFFT with an HRF neuron, we embed frequency selectivity directly into neuron dynamics.
> Despite removing spectral transforms, our model achieves comparable performance (~0.003). Gains are stronger on low-dimensional, long-sequence datasets with dominant per-variable temporal structure.
>
> In contrast, high-dimensional datasets (e.g., Electricity, Traffic) benefit from spectral methods due to stronger cross-variable dependencies and global context. Exchange shows limited gains due to its low dimensionality and short length.
>
> Overall, SpikF leverages global dependencies, while our HRF-based model provides a simpler alternative for structured temporal regimes.

---

> > ### Author Rebuttal · Reviewer_reh3 · 2026-04-03
> >
> > Thank you for the detailed and thoughtful rebuttal. I will maintain my original score 4.

---

> > > ### Author Response · Authors · 2026-04-03
> > >
> > > Thank you for your careful consideration and for engaging with our rebuttal! We really appreciate your feedback.
> > >
> > > We kindly encourage you to review the additional insights we provided in our second round of rebuttal to the other reviewers. We clarify the position of our work and state the various applications it can create an impact on [yV5Y]. Further, we summarise the benefits of heterogeneity in RNN and SSM setups [5dJF]. Next, we further clarified the benefits offered by the HRF dynamics and how discretisations can instil long-sequence modelling capabilities[rv2U].
> > >
> > >
> > > We hope this further clarifies the strength of this approach.
> > >
> > > edit-
> > >
> > > ---
> > > >Figure 6 suffers from poor readability due to overly small dimensions; key details of pre/post-HRF spike patterns and learned frequencies are obscured. Figure 1 has an unappealing and poorly formatted legend; visual elements and notation explanations lack clarity and aesthetic consistency.
> > >
> > > Thank you for your feedback on Figure clarity. We will ensure to update Figures 1 & 6 for clarity as suggested in the manuscript. Please let us know of any other such improvements.
> > >
> > > ---

---

### Decision · Program_Chairs · 2026-04-30

**Decision:**

Accept (regular)

**Comment:**

This paper proposes SH²RFSSM, a second-order Spiking State Space Model (SSM) that utilizes Harmonic Resonate-and-Fire (HRF) neurons for ultra-long time series modeling, including classification, regression, and multivariate forecasting. By integrating stable discretization schemes (IM/IMEX) and neuronal parameter heterogeneity, the proposed architecture eliminates the need for dense matrix multiplications, heavy non-linearities (like GeLU/GLU), and FFT-based signal processing. The result is a biologically grounded, fully spike-based model that achieves highly competitive performance against state-of-the-art non-spiking SSMs (e.g., LinOSS) and Transformers, while demonstrating massive theoretical energy efficiency improvements (e.g., ~69x energy reduction).

## Summary of the Review Process & Rebuttal
The paper initially received borderline to positive scores (three Weak Accepts, one Weak Reject). The reviewers universally praised the extensive empirical validation across 17 diverse datasets, the biological motivation of bridging SNNs with SSMs, and the impressive energy efficiency claims. However, reviewers raised several valid concerns:
1. **Motivation for Ultra-Long Sequences:** Reviewer rv2U questioned the theoretical depth of why HRF neurons are specifically suited for ultra-long dependencies compared to existing spiking SSMs.
2. **Impact of Heterogeneity:** Reviewer 5dJF and yV5Y found the initial heterogeneity ablation results marginal and confusing, questioning if heterogeneity genuinely contributed to the model's success.
3. **Energy Computation:** Reviewer 5dJF and rv2U questioned the validity of the energy consumption metrics, asking for details on the estimation method and whether real hardware validation was necessary.
4. **Model Distinctions & Baselines:** Reviewers asked for clearer distinctions between the SH²RFSSM and SpikHRFSSM variants, and explanations for baseline choices.

The authors provided a highly detailed, multi-round rebuttal that effectively resolved these issues:
* **Theoretical Grounding:** The authors supplied a rigorous mathematical proof (Theorem on Scaled Equivalence) demonstrating that undamped HRF neurons under IM/IMEX discretizations preserve information in the state's phase and amplitude. Unlike standard RNNs/SSMs where hidden states decay exponentially, the eigenvalues here remain on or near the unit circle, preventing signal vanishing and enabling stable ultra-long-range gradient flow.
* **Heterogeneity Clarification:** The authors decoupled their ablation tables to present a much clearer picture. By comparing a strictly homogeneous baseline (u, v, b = 0) against their heterogeneous model, they demonstrated a significant performance jump (e.g., from an average of 61.7% to 67.5% across UEA datasets), firmly validating the necessity of neuronal heterogeneity.
* **Energy Metrics & Hardware:** The authors clarified that their energy estimations use widely accepted 45nm CMOS hardware models, which is the standard evaluation protocol in the deep SNN literature.
* **Ablations & Clarifications:** Additional ablations on spike encoder normalizations (BN/LN vs. sequence length) and kernel sizes for regression were provided, alongside a clear distinction that SH²RFSSM is the core architecture while SpikHRFSSM is its specific application to replace computationally expensive FFTs in forecasting tasks.

## Meta Recommendation
The authors have done an excellent job addressing the reviewers' concerns during the rebuttal phase. Reviewers reh3, yV5Y, and rv2U acknowledged that their concerns were either fully or largely resolved, with yV5Y explicitly raising their evaluation. While Reviewer 5dJF maintained reservations regarding the lack of physical hardware measurements, the AC agrees with the authors that standardized hardware estimation (45nm CMOS) is more than sufficient for an algorithmic and architectural SNN paper at this venue.

The intersection of State Space Models and Spiking Neural Networks is a highly relevant and rapidly emerging research direction. This paper makes a substantial and original contribution to this space by moving beyond standard LIF neurons to more expressive second-order HRF neurons. The breadth of the experiments—spanning classification, human activity recognition, and 50k-length regression—proves the model's versatility. The theoretical insights into discretization and the empirical proofs of energy efficiency make this a strong paper that will undoubtedly inspire future work in low-power, edge AI sequence modeling.

The paper is technically solid, the rebuttal was exceptionally strong, and the minor presentation issues can easily be fixed in the camera-ready version. I recommend an Accept.